# Unleashing the Power of Annotation: Enhancing Semi-Supervised Learning through Unsupervised Sample Selection

## Abstract

With large volumes of unlabeled data and limited annotation budgets, Semi-Supervised Learning (SSL) has become a preferred approach in many deep learning tasks. However, most previous studies have primarily focused on utilizing labeled and unlabeled data for model training to improve performance, while the efficient selection of samples for annotation under budgetary constraints has often been overlooked.To fill this gap, we propose an efficient sample selection methodology named Unleashing the Power of Annotation (UPA). By adopting a modified Frank-Wolfe algorithm to minimizing a novel criterion $\alpha$-Maximum Mean Discrepancy ($\alpha$-MMD), UPA selects a representative and diverse subset for annotation from the unlabeled data. Furthermore, we demonstrate that minimizing $\alpha$-MMD enhances the generalization ability of low-budget learning. Experiments show that UPA consistently improves the performance of several popular SSL methods, surpassing various prevailing Active Learning (AL) and Semi-Supervised Active Learning (SSAL) methods even under constrained annotation budgets.

## 1 Introduction

Recent years have witnessed the great success of deep learning for a variety of supervised tasks, including image classification (Krizhevsky et al., 2012; He et al., 2016; Dosovitskiy et al., 2020; Liu et al., 2021a). However, these advances often rely heavily on large sets of high-quality labeled data. Unfortunately obtaining such data, especially in areas like finance, healthcare, and education where expert labeling is needed, can be both costly and challenging. To alleviate this hindrance, more research started to focus on Semi-Supervised Learning (SSL) (Berthelot et al., 2019b; Xie et al., 2020a; Zhang et al., 2021), which improves model performance with the help of large amounts of unlabeled data.

In semi-supervised learning, both the knowledge within unlabeled data and the supervision signals from labeled data are essential for its effectiveness (Xie et al., 2020a; Wang et al., 2022d). And our empirical evidence (see Sec. 4.3) shows that selecting labeled data significantly affects the final results, especially with very limited annotation budgets. Therefore, choosing which sample to label is crucial in the realm of SSL.

Previous SSL approaches have employed two common strategies for sample selection (Berthelot et al., 2019b;a; Wang et al., 2022b). One is random sampling from the entire dataset which may introduce imbalanced class distributions and inadequate coverage of the overall data distribution, resulting in poor performance. The other is random sampling within each class (*a.k.a.* stratified sampling), but it is not practical in real-world scenarios where label for each sample is unknown.

Semi-Supervised Active Learning (SSAL) seems to be a potential solution to the above issues by selecting the most suitable samples for annotation in each iteration of training inspired by Active Learning (AL) algorithms (Sener & Savarese, 2018). It also benefits significantly from unlabeled data, which in turn enhances the model's learning capability and prediction accuracy (Wang et al., 2022b; Guo et al., 2021). Nevertheless, analogous to AL methods, SSAL methods rely on initially randomly labeled data and require multiple rounds of sample selection for labeling, which may

potentially introduce harmful bias into the selection process and increase annotation budget (Xie et al., 2023).

In order to address the above issues, we propose an efficient sampling method named Unleashing the Power of Annotation (UPA) that requests annotations only once and is decoupled from the downstream tasks. Specifically, inspired by the concept of Maximum Mean Discrepancy (MMD) (Gretton et al., 2006), we design a novel criterion named $\alpha$-MMD that is to be minimized to strikes a balance between the representativeness and diversity via a trade-off parameter $\alpha$, which ensures that the selected data distributes similarly with the entire unlabeled dataset and is not excessively concentrated. By using a modified Frank-Wolfe algorithm called Generalized Kernel Herding without Replacement (GKHR), we can get an efficient approximate solution to this minimization problem.

We prove that under certain Reproducing Kernel Hilbert Space (RKHS) assumption, $\alpha$-MMD bounds the difference between training with a low versus an unlimited labeling budget, implying that our method could theoretically enhance the generalization ability of low-budget learning. We also give a theoretical assessment for GKHR with some supplementary numerical experiments, showing that GKHR performs well under low-budget setting. Additionally, we find an optimal interval for the value of the trade-off parameter $\alpha$ to ensure that the selected data remains representativeness.

Furthermore, we benchmark on our sample selection with several popular SSL frameworks on the public datasets CIFAR-10/100 (Krizhevsky et al., 2009) and SVHN (Netzer et al., 2011). Extensive experiments show that UPA outperforms other sample selection methods across different SSL frameworks. Remarkably, even with a constrained annotation budget, UPA demonstrates competitive performance against established AL and SSAL methods when applied to the SSL frameworks.

The main contributions of this article are as follows: (1) We propose UPA, a methodology for unsupervised sample selection, by minimizing a novel criterion $\alpha$-MMD which evaluates representativeness and diversity of selected samples. Under low-budget setting, we develop a fast and efficient algorithm GKHR for optimization. (2) We prove that our method benefits the generalizability of the trained model under certain assumptions, and rigorously establish an optimal interval for the trade-off parameter $\alpha$ to guarantee the representativeness of selected data. (3) We conduct extensive benchmarking for UPA across several popular SSL frameworks. The results demonstrate superior sample efficiency compared to alternative sample selection strategies. Moreover, even under fewer annotation budgets, UPA outperforms widely used AL/SSAL approaches when applied to the SSL frameworks.

## 2 RELATED WORK

### 2.1 SEMI-SUPERVISED LEARNING

Semi-Supervised Learning effectively utilizes sparse labeled data and abundant unlabeled data for model training. Consistency Regularization (Sajjadi et al., 2016; Laine & Aila, 2016; Tarvainen & Valpola, 2017; Xie et al., 2020a), Pseudo-Labeling (Lee et al., 2013; Xie et al., 2020b) and their hybrid strategies (Sohn et al., 2020; Zhang et al., 2021; Wang et al., 2022d) are commonly used in SSL. Consistency Regularization makes sure the model's output stays stable even when there's noise or small changes in the input, usually from the data augmentation. This approach fosters output consistency, as illustrated by Unsupervised Data Augmentation (UDA) (Xie et al., 2020a) deploying a single and weak augmentation. Pseudo-Labeling, adhering to entropy minimization, integrates high-confidence data pseudo-labels directly into training. Lee et al. (2013) adopt the maximum confidence prediction in a batch of unlabeled samples as a pseudo-label, which has the maximum predicted probability. Moreover, an integrative approach that combines the aforementioned strategies can also achieve substantial results, such as Fixmatch (Sohn et al., 2020), Flexmatch (Zhang et al., 2021) and Freematch (Wang et al., 2022d). Even though these techniques have proven effective, semi-supervised learning usually involves random label selection, which could hinder optimal results. Hence, we aim to create a more efficient strategy for sample selection, with the goal of enhancing these methods' performance.

## 2.2 ACTIVE LEARNING

Active learning (AL) aims to optimize the learning process by selecting informative samples for labeling, reducing reliance on large labeled datasets. There are two different criteria for sample selection: uncertainty and diversity. Uncertainty sampling, selects samples about which the current model is most uncertain. Earlier studies utilized posterior probability (Lewis & Catlett, 1994; Wang et al., 2016), entropy (Joshi et al., 2009; Luo et al., 2013), and classification margin (Tong & Koller, 2001) to estimate uncertainty. Recent research estimates uncertainty by calculating training loss directly (Huang et al., 2021; Yoo & Kweon, 2019) or evaluating influence on model performance (Freytag et al., 2014; Liu et al., 2021b). However, uncertainty sampling methods may exhibit performance disparities across different models, leading researchers to focus on diversity sampling which aims to align the distribution of selected samples with that of unlabeled ones. For instance, Sener & Savarese (2018) formulate the sample selection process from a theoretical standpoint, casting it as a $k$-Center problem, leading to the proposal of the CoreSet algorithm. Sinha et al. (2019) harness the power of adversarial networks to discern between labeled and unlabeled samples. Mahmood et al. (2021) solve the sample selection problem by Generalized Benders Decomposition algorithm which minimizes the Wasserstein distance between selected samples and unlabeled pool. Cho et al. (2022) select the most informative samples by maximizing the prediction difference across multiple classifiers. Contrasting with these aforementioned AL methods, which typically necessitate iterative rounds of sample selection tightly coupled with model training, UPA stands out as a model-agnostic algorithm. Notably, it streamlines the process by ensuring annotations are selected in a singular step.

## 3 METHODOLOGY

In this section, we will introduce our method on the unsupervised sample selection task. Firstly we formulate the task and the concept of representativeness and diversity in Sec. 3.1. Then we quantify the representativeness and diversity by only one criterion called $\alpha$-Maximum Mean Discrepancy ($\alpha$-MMD) in Sec. 3.2. Next, we propose a sampling algorithm called Generalized Kernel Herding without Replacement (GKHR) for our task, and conduct a brief study on its theoretical properties and empirical performance in Sec. 3.3. Finally, we introduce our strategies for choosing a proper kernel and give a optimal interval for parameter $\alpha$ to preserve the representativeness of selected data in Sec. 3.4.

### 3.1 FORMULATION OF UNSUPERVISED SAMPLE SELECTION TASK

Let $\mathcal{X}$ be the unlabeled data space, $\mathcal{Y}$ be the label space, $\mathcal{H}$ be a hypothesis set which consists of hypotheses $h : \mathcal{X} \rightarrow \mathcal{Y}$, and $\mathbf{X}_n = \{\mathbf{x}_i\}_{i \in [n]} \subset \mathcal{X}$ is the full unlabeled dataset. In order to reduce the labelling budget, there is a sampling algorithm $\mathcal{A}$ that selects $m \leq n$ unlabeled data $\{\mathbf{x}_i\}_{i \in \mathcal{I}_m}$ from $\mathbf{X}_n$, where $|\mathcal{I}_m| = m$ is an index set that is contained in $[n]$. The selected samples are denoted by $\mathbf{X}_{\mathcal{I}_m} = \{\mathbf{x}_i\}_{i \in \mathcal{I}_m}$. Once the selected samples $\mathbf{X}_{\mathcal{I}_m}$ are obtained, we can get access to the true labels of them, and then derive a set of training data $S = \{(\mathbf{x}_i, y_i)\}_{i \in \mathcal{I}_m}$. We can finally find a hypothesis $h_S$ from hypothesis set $\mathcal{H}$ using training data $S$.

We propose two fundamental settings for our task before introducing the main goal. The first one is the low-budget setting, we only consider the case when $m/n \leq 0.2$ in the following context, including the numerical study on the sampling algorithm and the experiments. The second one is the without-replacement setting, since unsupervised sample selection task is always regarded as a combinatorial optimization problem (Sener & Savarese, 2018; Mahmood et al., 2021; Wang et al., 2022a), where we must ensure that the selected samples are different from each other.

In our task, the sampling algorithm $\mathcal{A}$ is blind to the hypothesis set $\mathcal{H}$, and therefore it is a model-agnostic algorithm. Our main goal is to find a sampling strategy associate with a algorithm $\mathcal{A}$ such that the selected samples enjoy both representativeness and diversity, which are two fundamental criteria we employ here. Representativeness is designed to ensure that the selected samples distribute similarly with the entire unlabeled dataset. In contrast, diversity is critical in preventing an excessive concentration of selected samples in high-density areas of the original instance set, thus promoting an inclusive representation across the entire data landscape.

### 3.2 Quantification of Representativeness and Diversity

To quantify the representativeness of $\mathbf{X}_{\mathcal{I}_m}$ to $\mathbf{X}_n$, we may introduce a metric to measure the distance between them, where larger distance implies lower representativeness. In this paper, we use the Maximum Mean Discrepancy (MMD) for measuring the representativeness of selected samples. Proposed by (Gretton et al., 2006), MMD is formally defined below:

**Definition 1.** *Let $P, Q$ be two Borel probability measure on $\mathbb{R}^d$. Suppose $f$ is sampled from the unit ball in a reproducing kernel Hilbert space (RKHS) $\mathcal{H}$ associated with its reproducing kernel $k(\cdot, \cdot)$, i.e., $\|f\|_{\mathcal{H}} \leq 1$, then the MMD between $P$ and $Q$ is defined by*

$$
\begin{aligned}
\mathrm{MMD}_k^2(P, Q) &:= \sup_{\|f\|_{\mathcal{H}} \leq 1} \left( \int f dP - \int f dQ \right)^2 \\
&= \mathbb{E}\left[ k\left( X, X' \right) + k\left( Y, Y' \right) - 2k(X, Y) \right]
\end{aligned}
\tag{1}
$$

*where $X, X' \sim P$ and $Y, Y' \sim Q$ are independent copies.*

We can derive the empirical version of MMD that is able to measure the distance between $\mathbf{X}_{\mathcal{I}_m} = \{\mathbf{x}_i\}_{i \in \mathcal{I}_m}$ and $\mathbf{X}_n = \{\mathbf{x}_i\}_{i=1}^n$ if we substitute $P, Q$ by the empirical distribution constructed by $\mathbf{X}_{\mathcal{I}_m}, \mathbf{X}_n$ in (1):

$$
\mathrm{MMD}_k^2(\mathbf{X}_{\mathcal{I}_m}, \mathbf{X}_n) := \frac{1}{n^2} \sum_{i=1}^n \sum_{j=1}^n k\left( \mathbf{x}_i, \mathbf{x}_j \right) + \frac{1}{m^2} \sum_{i \in \mathcal{I}_m} \sum_{j \in \mathcal{I}_m} k\left( \mathbf{x}_i, \mathbf{x}_j \right) - \frac{2}{mn} \sum_{i=1}^n \sum_{j \in \mathcal{I}_m} k\left( \mathbf{x}_i, \mathbf{x}_j \right)
\tag{2}
$$

When $k$ is a characteristic kernel, i.e., $\mu \to \int_{\mathcal{X}} k(\cdot, \mathbf{x}) d\mu(\mathbf{x})$ for any Borel probability measure $\mu$ on $\mathcal{X}$, MMD becomes a proper metric in the space of all Borel probability measure (Muandet et al., 2017), making itself a good choice for measuring the representativeness. Moreover, if the kernel $k$ is a radial kernel with $k(\mathbf{x}, \mathbf{y}) = \psi(\|\mathbf{x} - \mathbf{y}\|)$ where $\psi$ is a bounded decreasing function on $\mathbb{R}_+$, $k$ could be regarded as a criterion of similarity between two samples. Additionally, $k$ should be positive definite so that it corresponds to a unique RKHS. Therefore, in the following context, we assume the kernel $k$ is positive definite, characteristic and radial.

Define

$$
D(\mathbf{X}_{\mathcal{I}_m}) = \frac{1}{m^2} \sum_{i \in \mathcal{I}_m} \sum_{j \in \mathcal{I}_m} k\left( \mathbf{x}_i, \mathbf{x}_j \right), D(\mathbf{X}_{\mathcal{I}_m}, \mathbf{X}_n) = \frac{1}{mn} \sum_{i=1}^n \sum_{j \in \mathcal{I}_m} k\left( \mathbf{x}_i, \mathbf{x}_j \right)
$$

then regarding $k$ as a similarity criterion, $D(\mathbf{X}_{\mathcal{I}_m})$ evaluates the self-similarity of $\mathbf{X}_{\mathcal{I}_m}$, and $D(\mathbf{X}_{\mathcal{I}_m}, \mathbf{X}_n)$ evaluates the similarity between $\mathbf{X}_{\mathcal{I}_m}$ and $\mathbf{X}_n$, while the former one can be interpreted as the quantification of diversity of selected samples. Since (2) can be rewritten by

$$
\mathrm{MMD}_k^2(\mathbf{X}_{\mathcal{I}_m}, \mathbf{X}_n) = D(\mathbf{X}_n) + D(\mathbf{X}_{\mathcal{I}_m}) - 2D(\mathbf{X}_{\mathcal{I}_m}, \mathbf{X}_n)
$$

then MMD is indeed a function of $\mathbf{X}_{\mathcal{I}}$ that contains a diversity term. Thus we can define a new concept called the $\alpha$-MMD by reweighting each term in (1),(2) to measure the both representativeness and diversity of selected data, and the preference for the diversity of selected samples is determined by a single parameter $\alpha$.

**Definition 2.** *Following the settings in Definition 1, the $\alpha$-MMD between distribution $P$ and $Q$ is defined by*

$$
\mathrm{MMD}_{k,\alpha}^2(P, Q) := \sup_{\|f\|_{\mathcal{H}} \leq 1} \left( \int f dP - \alpha \int f dQ \right)^2 = \mathbb{E}\left[ k\left( X, X' \right) + \alpha^2 k\left( Y, Y' \right) - 2\alpha k(X, Y) \right]
\tag{3}
$$

*and for $\mathbf{X}_{\mathcal{I}_m} = \{\mathbf{x}_i\}_{i \in \mathcal{I}_m}$ and $\mathbf{X}_n = \{\mathbf{x}_i\}_{i=1}^n$,*

$$
\mathrm{MMD}_{k,\alpha}^2(\mathbf{X}_{\mathcal{I}_m}, \mathbf{X}_n) := \frac{\alpha^2}{n^2} \sum_{i=1}^n \sum_{j=1}^n k\left( \mathbf{x}_i, \mathbf{x}_j \right) + \frac{1}{m^2} \sum_{i \in \mathcal{I}_m} \sum_{j \in \mathcal{I}_m} k\left( \mathbf{x}_i, \mathbf{x}_j \right) - \frac{2\alpha}{mn} \sum_{i=1}^n \sum_{j \in \mathcal{I}_m} k\left( \mathbf{x}_i, \mathbf{x}_j \right)
\tag{4}
$$

In the following, we provide an explanation for the capability of $\alpha$-MMD to quantify representativeness and diversity from the perspective of regularization. By definition, minimizing $\alpha$-MMD is equivalent to minimizing

$$\mathcal{E}_\alpha(\mathbf{X}_{\mathcal{I}_m}, \mathbf{X}_n) := \underbrace{D(\mathbf{X}_{\mathcal{I}_m}) - 2D(\mathbf{X}_{\mathcal{I}_m}, \mathbf{X}_n)}_{(*)} + \underbrace{\left(\frac{1}{a} - 1\right) D(\mathbf{X}_{\mathcal{I}_m})}_{(**)} \tag{5}$$

where $(*)$ is the objective function for minimizing MMD, $(**)$ is the diversity regularizer with a parameter $\alpha$ controls the trade-off between representativeness and diversity in the regularization. To be more specific, larger value of $\alpha$ implies larger punishment on the diversity. Therefore, minimizing the $\alpha$-MMD between the selected samples and the full dataset where $\alpha \leq 1$ is a proper strategy to preserve the representativeness and diversity.

Recall the core-set approach in Sener & Savarese (2018), i.e., for any $h \in \mathcal{H}$,

$$R(h) \leq \widehat{R}_S(h) + |R(h) - \widehat{R}_T(h)| + |\widehat{R}_T(h) - \widehat{R}_S(h)|$$

where $T$ is the full labeled dataset and $S \subset T$ is the core set, $R(h)$ is the expected risk of $h$, $\widehat{R}_T(h), \widehat{R}_S(h)$ are empirical risk of $h$ on $T, S$. The first term $\widehat{R}_S(h)$ is unknown before we label that selected samples, the second term $|R(h) - \widehat{R}_T(h)|$ is regarded as the generalization bound which does not depend on the core set, therefore we only need to minimize the third term $|\widehat{R}_T(h) - \widehat{R}_S(h)|$ called core-set loss to control the upper bound of $R(h)$. Actually, the core-set loss can be upper bounded by $\alpha$-MMD under certain assumptions, implying that our approach could benefit the generalizability of the trained model.

**Theorem 3.1.** *Let $\mathcal{H}_1$ be a hypothesis set that contains some hypotheses $h : \mathcal{X} \to \mathcal{Y}$, and let the loss function $\ell$ be squared loss, i.e., $\ell(h(\mathbf{x}), y) = (h(\mathbf{x}) - y)^2$. Assume that the labelled data $T = \{(\mathbf{x}_i, y_i)\}_{i=1}^n$ are i.i.d. sampled from a random vector $(X, Y)$ defined on $\mathcal{X} \times \mathcal{Y}$. Let $\mathcal{H}_2, \mathcal{H}_3$ be two RKHS containing real-valued functions on $\mathcal{X}$, which are associated with bounded positive definite kernel $k_2, k_3$, further assume $\mathbb{E}(Y|X) \in \mathcal{H}_2$ with norm bounded by $K_m$, $\mathrm{Var}(Y|X) \in \mathcal{H}_3$ with norm bounded by $K_s$, and $\mathcal{H}_1$ is a RKHS associated with bounded positive definite kernel $k_1$ where the norm of any $h \in \mathcal{H}_1$ is bounded by $K_h$, then for any selected samples $S \subset T$, there exists a positive constant $K_c$ such that the following inequality holds*

$$|\widehat{R}_T(h) - \widehat{R}_S(h)| \leq K_c(\mathrm{MMD}_{k,\alpha}(\mathbf{X}_S, \mathbf{X}_T) + (1 - \alpha)\sqrt{K})^2$$

*where $\alpha \leq 1$, $0 \leq \max_{\mathbf{x} \in \mathcal{X}} k(\mathbf{x}, \mathbf{x}) \leq K$, $k = k_1^2 + k_1 k_2 + k_3$, $\mathbf{X}_S, \mathbf{X}_T$ are projections of $S, T$ on $\mathcal{X}$.*

**Remark 1.** For the RKHS assumptions of conditional expectation and conditional variance in Theorem 3.1, they are standard in the literature of embedding conditional distributions (Song et al., 2009; Sriperumbudur et al., 2012). For the RKHS assumption on $\mathcal{H}_1$, it is indeed applicable in many machine learning problem, such as SVM, kernel ridge regression and CNN (Bietti & Mairal, 2019). In general case, Theorem 3.1 does not always holds, nevertheless, since we are blind to the hypothesis set $\mathcal{H}$ before training, it can yet be regarded as a guideline for selecting samples.

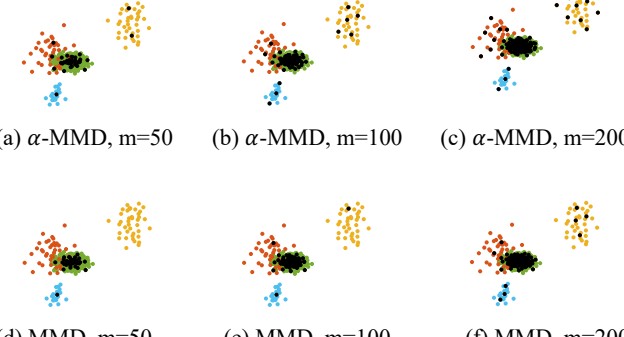

(a) $\alpha$-MMD, m=50    (b) $\alpha$-MMD, m=100    (c) $\alpha$-MMD, m=200

(d) MMD, m=50    (e) MMD, m=100    (f) MMD, m=200

Figure 1: Visualization of samples selected by minimizing $\alpha$-MMD and MMD, from a 2000-sample-dataset generated by Gaussian Mixture Model 1 (Appendix A.4).

## 3.3 SAMPLING ALGORITHM

In the previous research (Sener & Savarese, 2018; Mahmood et al., 2021; Wang et al., 2022a), selecting unsupervised samples is usually modeled by a combinatorial optimization problem which

is NP-hard. Follow the idea of Bach et al. (2012), we exploit the convexity of $\alpha$-MMD, regard $\min_{\mathcal{I}_m \in [n]} \mathcal{E}_\alpha(\mathbf{X}_{\mathcal{I}_m}, \mathbf{X}_n)$ as a continuous convex optimization problem and conduct a iterative minimization procedure by Frank-Wolfe algorithm. After rigorous derivation in Appendix A.1, we obtain the following iterating formula:

$$\mathcal{I}_0 = \emptyset, p = 0, \mathbf{x}_{i_{p+1}} \in \arg\min_{i \in [n]} f_{\mathcal{I}_p}(\mathbf{x}_i), \mathcal{I}_{p+1} \leftarrow \mathcal{I}_p \cup \{i_{p+1}\}, p \leftarrow p + 1 \tag{6}$$

where $f_{\mathcal{I}_p}(\mathbf{x}_i) = \sum_{j \in \mathcal{I}_p} k(\mathbf{x}, \mathbf{x}_j) - \alpha p \sum_{l=1}^n k(\mathbf{x}, \mathbf{x}_l)/n$. Actually, (6) is an extension of kernel herding in Chen et al. (2012), thus we name the corresponding algorithms of (6) by Generalized Kernel Herding (GKH), see Algorithm 1. However, same samples could be selected for several times in GKH, which contradicts the without-replacement setting. Hence, we modify GKH to a new algorithm called Generalized Kernel Herding without Replacement (GKHR), which selects new samples besides the selected samples $\mathbf{X}_{\mathcal{I}_m}$, see Algorithm 2. Its iterating formula is given below:

$$\mathcal{I}_0 = \emptyset, p = 0, \mathbf{x}_{i_{p+1}} \in \arg\min_{i \in [n] \setminus \mathcal{I}_p} f_{\mathcal{I}_p}(\mathbf{x}_i), \mathcal{I}_{p+1} \leftarrow \mathcal{I}_p \cup \{i_{p+1}\}, p \leftarrow p + 1 \tag{7}$$

Clearly, (7) admits no repetitiveness of the elements in selected samples.

Some theoretical and empirical assessments on the performance GKHR are conduct in the following. Firstly, the computational complexity of GKHR grows linearly and is $\mathcal{O}(nm)$ for $m$ iterations, since the $\sum_{j \in \mathcal{I}_m} k(\mathbf{x}, \mathbf{x}_j)$ term can be iteratively updated (details see Algorithm 2), reducing the complexity of each iteration to $O(n)$. Hence, GKHR is a fast algorithm for selecting selected samples.

As for the convergence property, actually, it is not applicable for GKHR. With $n$ fixed, GKHR iterates for at most $n$ times and when $m = n$, GKHR has a fixed output where $\mathbf{X}_{\mathcal{I}_n} = \mathbf{X}_n$ and $\mathrm{MMD}^2_{k,\alpha}(\mathbf{X}_{\mathcal{I}_n}, \mathbf{X}_n) = C_\alpha^2$. Nevertheless, we have the following theorem to show that the optimization error of GKHR is upper bounded when $m$ is sufficiently small.

**Theorem 3.2.** *Let $\mathbf{X}_{\mathcal{I}_m}$ be the samples selected by GKHR, then under the following assumption: for any $\mathcal{I}_p$, $1 \leq p \leq m - 1$, there exists $p + 1$ elements $\{\mathbf{x}_{j_l}\}_{l=1}^{p+1}$ in $\mathbf{X}_n$ such that*

$$f_{\mathcal{I}_p}(\mathbf{x}_{j_1}) \leq \cdots f_{\mathcal{I}_p}(\mathbf{x}_{j_{p+1}}) \leq \frac{\sum_{i=1}^n f_{\mathcal{I}_p}(\mathbf{x}_i)}{n}$$

*it holds that*

$$\mathrm{MMD}^2_{k,\alpha}(\mathbf{X}_{\mathcal{I}_m}, \mathbf{X}_n) \leq C_\alpha^2 + B\frac{2 + \log m}{m + 1} \tag{8}$$

*where $B = 2K$, $0 \leq \max_{\mathbf{x} \in \mathcal{X}} k(\mathbf{x}, \mathbf{x}) \leq K$, $C_\alpha^2 = (1 - \alpha)^2 \overline{K}$ where $\overline{K}$ is defined in Lemma A.6.*

**Remark 2.** The assumption is actually an extension of the principle of "the minimum is never larger than the mean". In the unsupervised sample selection task where there is a low-budget setting such that $m/n \leq 0.2$, the assumption still makes sense.

To ensure that GKHR indeed works in our unsupervised data selection task, we conduct some numerical experiments to empirically compare the performance of GKHR with GKH on several datasets generated by predefined distributions, since GKH is a convergent algorithm and the finite-sample-size error bound (9) holds without any assumption on the data. The results in Appendix A.4 show that GKHR and GKH have similar empirical performance on minimizing $\alpha$-MMD under low-budget setting, which convince us that GKHR could perform well in the unsupervised sample selection task.

### 3.4 CHOICE OF KERNEL $k$ AND PARAMETER $\alpha$

In Sec. 3.2, we claimed that we only consider positive definite, characteristic and radial kernels, among which Gaussian kernel (or RBF) is the most popular choice in the previous research. In this paper, we also adopt Gaussian kernel in the experiments. Therefore, another problem arises, that is how to choose the bandwidth parameter $\sigma$. Here we set $\sigma$ to be the median distance between samples in the aggregate dataset which is recommended by Gretton et al. (2012), since median is robust and also compromises between extreme cases. Specifically, for Gaussian kernel $k(\mathbf{x}, \mathbf{y}) = \exp(\|\mathbf{x} - \mathbf{y}\|_2^2)/\sigma^2$, we set

$$\sigma = \mathrm{Median}(\{\|\mathbf{x} - \mathbf{y}\|_2 | \mathbf{x}, \mathbf{y} \in \mathbf{X}_n\})$$

As for choosing $\alpha$, according to Theorem 3.2 and Lemma A.3, by straightforward deduction we have

$$\mathrm{MMD}_k\left(\mathbf{X}_{\mathcal{I}_m}, \mathbf{X}_n\right) \leq C_\alpha + \mathcal{O}\left(\sqrt{\frac{\log m}{m}}\right) + (1-\alpha)\sqrt{K}$$

to upper bound the MMD between the selected samples and the full dataset in low-budget setting. We can just set $1 - \sqrt{1/m} \leq \alpha \leq 1$ so that the upper bound of the MMD would not be larger than the one of $\alpha$-MMD in the perspective of order of magnitude.

## 4 EXPERIMENTS

In this section, we first introduce the datasets used for experiments in Sec. 4.1. Then we explain the implementation details of our method UPA in Sec. 4.2. Next, we evaluate UPA by integrating them into two popular SSL methods (FlexMatch (Zhang et al., 2021) and Freematch (Wang et al., 2022d)) on three benchmarks (CIFAR-10/100 and SVHN) in Sec. 4.3. We also compare against various AL/SSAL methods in Sec. 4.4. Lastly, we make visualization and quantitative analyses of our method in Sec. 4.5.

### 4.1 DATASETS

We choose three common datasets CIFAR-10, CIFAR-100 and SVHN for experiments. CIFAR-10 and CIFAR-100 contain 60,000 images with 10 and 100 categories separately, among which 50,000 images are for training and 10,000 images are for testing; SVHN contains 73,257 images for training and 26,032 images for testing. The training sets of the above datasets are considered as the unlabeled dataset for sample selection.

### 4.2 IMPLEMENTATION DETAILS OF OUR METHOD

First, we leverage the pre-trained image feature extraction capabilities of CLIP (Radford et al., 2021), which is a vision transformer architecture, to extract features. Subsequently, the [CLS] token features produced by the model's final output are employed for sample selection. During the sample selection phase, the Gaussian kernel function is chosen as the kernel method to compute the similarity of samples in an infinite-dimensional feature space. The value of $\sigma$ for the Gaussian kernel function is set as explained in Sec. 3.4. To ensure diversity in the sampled data, we introduce a penalty factor given by $\alpha = 1 - \frac{1}{\sqrt{m}}$, where $m$ denotes the numbers of selected samples. Concretely, we set $m = \{40, 250, 4000\}$ for CIFAR-10, $m = \{400, 2500, 10000\}$ for CIFAR-100, and $m = \{250, 1000\}$ for SVHN. Next, the selected samples are used for two SSL methods which are trained and evaluated on the three datasets by using the Unified SSL Benchmark (USB) (Wang et al., 2022c). Specifically, we use Wide ResNet-28-2 (Zagoruyko & Komodakis, 2016) as the backbone of all SSL methods and SGD with a momentum of 0.9 as optimizer. The initial learning rate is 0.03 with a learning rate decay 0.0005 except for CIFAR-100 where we set 0.001. Finally, we evaluate the performance with the Top-1 classification accuracy metric on the test set. Experiments are run on 8*NVIDIA Tesla A100 (40 GB) and 2*Intel 6248R 24-Core Processor. We average our results over three independent runs. In order to ensure the reproducibility of the experiments, we set certain random seeds for all experiments.

### 4.3 COMPARISON WITH OTHER SAMPLING METHODS

In order to verify the effectiveness of our sampling method UPA, we apply UPA on Flexmatch and Freematch to compare with the following two sampling methods which are widely used in SSL tasks under different annotation budget conditions:

(1) **Random:** The samples for annotation are selected randomly over all the available data.

(2) **Stratified:** The samples for annotation are selected randomly over individual categories evenly.

We report the **mean and standard deviation** of results over three trials in Table 1 on which we have several observations: (1) From the perspective of annotation budget, UPA consistently outperforms random sampling across varying budget constraints. This superiority in performance is evident

Table 1: Comparison with other sampling methods. Each result shows mean accuracy and standard deviation over three independent runs. Due to limitations of stratified sampling, results of which are marked in gray. Numbers in parentheses show improvement over random sampling.

| Datasets | Budget | Flexmatch | | | Freematch | | |
|---|---|---|---|---|---|---|---|
| | | Stratified | Random | UPA (Ours) | Stratified | Random | UPA(Ours) |
| CIFAR-10 | 40 | 89.6±3.3 | 90.4±3.2 | **94.8±0.2** (↑ **4.4**) | 95.0±0.1 | 94.3±0.7 | **95.0±0.1** (↑ **0.7**) |
| | 250 | 95.3±0.1 | 94.5±0.4 | **95.1±0.5** (↑ **0.6**) | 95.5±0.3 | 94.5±0.7 | **95.6±0.1** (↑ **1.1**) |
| | 4000 | 95.6±0.1 | 95.5±0.2 | **95.7±0.1** (↑ **0.2**) | 95.6±0.1 | 95.5±0.1 | **95.9±0.3** (↑ **0.4**) |
| CIFAR-100 | 400 | 50.4±0.4 | 45.9±1.1 | **48.4±0.3** (↑ **2.5**) | 51.6±0.4 | 48.1±0.7 | **48.3±0.5** (↑ **0.2**) |
| | 2500 | 67.5±0.5 | 66.8±1.1 | **67.3±0.5** (↑ **0.5**) | 67.5±0.7 | 66.8±0.8 | **67.4±0.3** (↑ **0.6**) |
| | 10000 | 73.4±0.4 | 73.0±0.4 | **73.4±0.1** (↑ **0.4**) | 73.6±0.5 | 72.8±0.5 | **73.1±0.3** (↑ **0.3**) |
| SVHN | 250 | 88.6±1.8 | 88.0±1.3 | **91.6±0.5** (↑ **3.6**) | 92.8±1.3 | 92.2±2.1 | **94.5±0.4** (↑ **2.3**) |
| | 1000 | 93.5±0.4 | 94.9±0.6 | **95.8±0.3** (↑ **0.9**) | 94.7±0.7 | 95.4±0.5 | **96.0±0.3** (↑ **0.6**) |

even under low budget conditions, manifesting in significant accuracy gains. For instance, when employing UPA, Flexmatch achieves an accuracy improvement of 4.4% on the CIFAR-10 dataset with an annotation budget of 40, a 2.5% increase on the CIFAR-100 dataset with a budget of 400, and a 3.6% rise on the SVHN with a budget of 250. Similarly, Freematch registers an accuracy enhancement of 1.1% on CIFAR-10 with an annotation budget of 250, and 2.3% on SVHN with a budget of 250. These results underscore the efficacy of UPA in judiciously selecting samples, thereby reducing the associated costs of sample labeling. (2) From the perspective of the SSL framework, UPA demonstrates improvements over random sampling across the SSL frameworks. This highlights that UPA is model-agnostic, exhibiting notable portability and the proficiency to operate within diverse SSL frameworks. (3) In the majority of experiments, UPA either approaches or surpasses the performance of the stratified sampling method. Notably, the stratified sampling method is practically infeasible given that the category labels of the data are not known a priori. This shows that UPA is better suited for real-world challenges.

## 4.4 COMPARISON WITH AL/SSAL METHODS

We compare our method UPA against various recent AL/SSAL methods when applied to two SSL frameworks in terms of budget and accuracy on CIFAR-10. AL methods conclude VAAL (Sinha et al., 2019), UncertainGCN (Caramalau et al., 2021), CoreGCN (Caramalau et al., 2021) and MC-DAL (Cho et al., 2022), while SSAL methods conclude CBSSAL (Gao et al., 2020), TOD-Semi (Huang et al., 2021), REVIVAL (Guo et al., 2021) and ActiveFT (Xie et al., 2023). The experimental results are shown in Table 2 where the results of AL and SSAL are from Wang et al. (2022a) and Xie et al. (2023). According to the results, we have several observations: (1) AL methods often necessitate significantly larger labeling budgets, exceeding UPA by a factor of 125 or more. This is primarily because AL paradigms are solely dependent on labeled samples not only for classification but also for feature learning. (2) SSAL methods leverage unlabeled samples, resulting in enhanced label efficiency. While their performance modestly surpasses traditional AL methods, the improvement is not substantial. One plausible reason could be that, akin to AL methods, SSAL methods initialize with randomly labeled samples. This initialization might inadvertently inject detrimental biases during the early stages of the training procedure. (3) UPA demonstrates notable enhancements in accuracy while operating under a substantially reduced annotation budget. This highlights proficiency of UPA in exploiting unlabeled samples during the sample selection phase.

Table 2: Comparison with AL/SSAL methods on CIFAR-10.

| Methods | Budget | Acc(%) |
|---|---|---|
| *Active Learning (AL)* | | |
| VAAL | 7500 | 86.8 |
| UncertainGCN | 7500 | 86.8 |
| CoreGCN | 7500 | 86.5 |
| MCDAL | 7500 | 87.2 |
| *Semi-Supervised Active Learning (SAAL)* | | |
| CBSSAL | 150 | 87.6 |
| TOD-Semi | 7500 | 87.8 |
| REVIVAL | 150 | 88.0 |
| ActiveFT | 500 | 88.2 |
| *Semi-Supervised Learning (SSL) with UPA* | | |
| FlexMatch+UPA (Ours) | **40** | 94.8 |
| FreeMatch+UPA (Ours) | **40** | **95.0** |

## 4.5 ANALYSIS

### 4.5.1 VISUALIZATION OF SELECTED SAMPLES

To offer a more intuitive comparison between various sampling methods, we visualized samples chosen by random, stratified and our methods. We generate 5000 samples from a Gaussian mixture model defined on $\mathbb{R}^2$ with 10 components and uniform mixture weights. One hundred samples are selected from the entire dataset using different sampling methods. The results of visualization in Figure 2 indicate that: (1) Samples selected by stratified sampling method are more evenly distributed across different classes compared to the ones selected by random sampling method, given its explicit reliance on labels. (2) Samples selected by UPA also achieve uniform distribution across classes, which guarantees comprehensive coverage of the entire dataset. Simultaneously, due to the introduction of a novel diversity assessment criterion in our method, the selected samples do not overly cluster within a class, thus avoiding label redundancy.

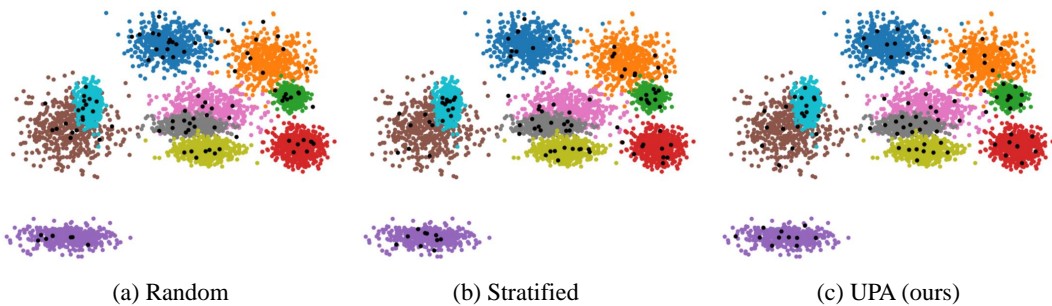

|        (a) Random        |        (b) Stratified        |        (c) UPA (ours)        |

Figure 2: Visualization of selected samples using different sampling methods. Points of different colors represent samples from different classes, while black points indicate the selected samples.

### 4.5.2 TRADE-OFF PARAMETER $\alpha$

We analyze the effect of different trade-off parameter $\alpha$ with Freematch on CIFAR-100. Table 3 shows the results of different $\alpha$ where the gray result indicates that the $\alpha$ used in this experiment is outside the interval we set in Sec. 3.4, *i.e.* $\alpha < 1 - \frac{1}{\sqrt{m}}$, while the black result indicates that the $\alpha$ used in this experiment is within the interval we set, *i.e.* $1 - \frac{1}{\sqrt{m}} \leq \alpha \leq 1$. From Table 3, we can find that: (1) The results of $\alpha$ within the interval are almost better than those outside the interval; (2) The $\alpha$ that achieve the best performance under different annotation budgets are within the interval we set. The above experimental results are in line with our theoretical derivation in Sec. 3.4.

Table 3: Effect of different $\alpha$ on CIFAR-100. The gray result indicates that $\alpha$ is outside the interval we set, while the black result indicates that $\alpha$ is inside the interval.

| $\alpha$ | CIFAR-100 | | |
|---|---|---|---|
| | 400 | 2500 | 10000 |
| 0.80 | 45.7±0.3 | 66.8±0.2 | 72.6±0.1 |
| 0.90 | 48.9±0.5 | 67.1±0.2 | 72.1±0.3 |
| 0.95 | 48.3±0.5 | 67.0±0.3 | 72.5±0.2 |
| 0.98 | **49.0±0.4** | **67.4±0.3** | 72.6±0.3 |
| 1.00 | 48.6±0.5 | 67.0±0.2 | 73.0±0.1 |
| Ours | 48.3±0.5 | **67.4±0.3** | **73.1±0.3** |

## 5 CONCLUSION

In this work, we propose an efficient sampling method UPA to select a subset of data from unlabeled data for annotation in SSL. The primary innovation of our approach lies in the introduction of $\alpha$-MMD, designed to valuate representativeness and diversity of selected samples. Under low-budget setting, we develop a fast and efficient algorithm GKHR for this problem using Frank-Wolfe algorithm. Both theoretical analyses and empirical experiments validate the efficacy of UPA. Remarkably, UPA is a model-agnostic sampling method, necessitating annotations only on an initial instance, rendering it adaptable across different SSL frameworks and simultaneously reducing annotation budget. We hope that our contribution will draw more attention from the community, fostering a more expansive perspective on sample selection within SSL.

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

# A APPENDIX

## A.1 ALGORITHMS

*Derivation of Generalized Kernel Herding (GKH).* Let us firstly define a weighted version of $\alpha$-MMD. For any $\mathbf{w} \in \mathbb{R}^n$ such that $\mathbf{w}^\top \mathbf{1} = 1$,

$$\text{MMD}^2_{k,\alpha,\mathbf{X}_n}(\mathbf{w}) = \mathbf{w}^\top \mathbf{K} \mathbf{w} - 2\alpha \mathbf{w}^\top \mathbf{p} + \alpha^2 \overline{K}$$

where $\mathbf{K} = [k(\mathbf{x}_i, \mathbf{x}_j)]_{1 \le i,j \le n}$, $\overline{K} = \mathbf{1}^\top \mathbf{K} \mathbf{1}/n^2$, $\mathbf{p} = (\mathbf{e}_1^\top \mathbf{K} \mathbf{1}/n, \cdots, \mathbf{e}_n^\top \mathbf{K} \mathbf{1}/n)$, $\{\mathbf{e}_i\}_{i=1}^n$ is the set of standard basis of $\mathbb{R}^n$, we have for any $\mathcal{I}_p \subset [n]$,

$$\text{MMD}^2_{k,\alpha,\mathbf{X}_n}(\mathbf{w}_p) = \text{MMD}^2_{k,\alpha}(\mathbf{X}_{\mathcal{I}_p}, \mathbf{X}_n)$$

where $(\mathbf{w}_p)_i = 1/p$ if $i \in \mathcal{I}$, and $(\mathbf{w}_p)_i = 0$ if not. Therefore, weighted $\alpha$-MMD is indeed a generalization of $\alpha$-MMD. Let

$$\mathbf{K}_* = \mathbf{K} - 2\alpha \mathbf{p} \mathbf{1}^\top + \alpha^2 \overline{K} \mathbf{1} \mathbf{1}^\top$$

we obtain the quadratic form expression of generalized $\alpha$-MMD by $\text{MMD}^2_{k,\alpha,\mathbf{X}_n}(\mathbf{w}) = \mathbf{w}^\top \mathbf{K}_* \mathbf{w}$, where $\mathbf{K}_*$ is strictly positive definite if $\mathbf{w} \ne \mathbf{w}_n$ and $k$ is a characteristic kernel according to Pronzato (2021). Recall our low-budget setting and choice of kernel, $\mathbf{K}_*$ is indeed a strictly positive definite matrix. Thus $\text{MMD}^2_{k,\alpha,\mathbf{X}_n}$ is a convex functional w.r.t. $\mathbf{w}$, leading to the fact that $\min_{\mathbf{w}^\top \mathbf{1}=1} \text{MMD}^2_{k,\alpha,\mathbf{X}_n}(\mathbf{w})$ can be solved by Frank-Wolfe algorithm. Then for $1 \le p < n$,

$$\mathbf{s}_p \in \arg\min_{\mathbf{s}^\top \mathbf{1}=1} \mathbf{s}^\top (\mathbf{K}\mathbf{w}_p - \alpha \mathbf{p}) = \arg\min_{\mathbf{e}_i, i \in [n]} \mathbf{e}_i^\top (\mathbf{K}\mathbf{w}_p - \alpha \mathbf{p})$$

let $\mathbf{e}_{i_p} = \mathbf{s}_p$, under uniform step size, we have

$$\mathbf{w}_{p+1} = \left(\frac{p}{p+1}\right) \mathbf{w}_p + \frac{1}{p+1} \mathbf{e}_{i_p}$$

as the update formula of Frank-Wolfe algorithm, which is equivalent to

$$i_p \in \arg\min_{i \in [n]} \sum_{j \in \mathcal{I}_m} k(\mathbf{x}_i, \mathbf{x}_j) - \alpha p \sum_{l=1}^n k(\mathbf{x}_i, \mathbf{x}_l)$$

for $\mathbf{w}_1$, we set $\mathbf{w}_0 = 0$. Then we immediately derive the iterating formula in (6). $\qquad\square$

---

**Algorithm 1** Generalized Kernel Herding

---

**Input:** Data set $\mathbf{X}_n = \{\mathbf{x}_1, \cdots, \mathbf{x}_n\} \subset \mathcal{X}$; the number of selected samples $m < n$; a positive definite, characteristic and radial kernel $k(\cdot, \cdot)$ on $\mathcal{X} \times \mathcal{X}$; trade-off parameter $\alpha \le 1$.
**Output:** selected samples $\mathbf{X}_{\mathcal{I}_m} = \{\mathbf{x}_{i_1}, \cdots, \mathbf{x}_{i_m}\}$.
 1: For each $\mathbf{x}_i \in \mathbf{X}_n$ calculate $\mu(\mathbf{x}_i) := \sum_{j=1}^n k(\mathbf{x}_j, \mathbf{x}_i)/n$.
 2: Set $\beta_1 = 1, S_0 = 0, \mathcal{I} = \emptyset$.
 3: **for** $p \in \{1, \cdots, m\}$ **do**
 4: $\quad i_p \in \arg\min_{i \in [n]} S_{p-1}(\mathbf{x}_i) - \alpha \mu(\mathbf{x}_i)$
 5: $\quad$ For all $i \in [n]$, update $S_p(\mathbf{x}_i) = (1 - \beta_p) S_{p-1} + \beta_p k(\mathbf{x}_{i_p}, \mathbf{x})$
 6: $\quad \mathcal{I}_{p+1} \leftarrow \mathcal{I}_p \cup \{i_p\}, p \leftarrow p + 1$, set $\beta_p = 1/p$.
 7: **end for**

---

## A.2 TECHNICAL LEMMAS

**Lemma A.1** (Lemma 2, Pronzato (2021)). *Let $(t_k)_k$ and $(\alpha_k)_k$ be two real positive sequences and $A$ be a strictly positive real. If $t_k$ satisfies*

$$t_1 \le A \text{ and } t_{k+1} \le (1 - \alpha_{k+1}) t_k + A \alpha_{k+1}^2, k \ge 1,$$

*with $\alpha_k = 1/k$ for all $k$, then $t_k < A(2 + \log k)/(k+1)$ for all $k > 1$.*

---

**Algorithm 2** Kernel Herding without Replacement

---

**Input:** Data set $\mathbf{X}_n = \{\mathbf{x}_1, \cdots, \mathbf{x}_n\} \subset \mathcal{X}$; the number of selected samples $m < n$; a positive definite, characteristic and radial kernel $k(\cdot, \cdot)$ on $\mathcal{X} \times \mathcal{X}$; trade-off parameter $\alpha \leq 1$.
**Output:** selected samples $\mathbf{X}_{\mathcal{I}_m} = \{\mathbf{x}_{i_1}, \cdots, \mathbf{x}_{i_m}\}$.
 1: For each $\mathbf{x}_i \in \mathbf{X}_n$ calculate $\mu(\mathbf{x}_i) := \sum_{j=1}^{n} k(\mathbf{x}_j, \mathbf{x}_i)/n$.
 2: Set $\beta_1 = 1$, $S_0 = 0$, $\mathcal{I} = \emptyset$.
 3: **for** $p \in \{1, \cdots, m\}$ **do**
 4:     $i_p \in \arg\min_{i \in [n] \backslash \mathcal{I}_p} S_{p-1}(\mathbf{x}_i) - \alpha\mu(\mathbf{x}_i)$
 5:     For all $i \in [n] \backslash \mathcal{I}_p$, update $S_p(\mathbf{x}_i) = (1 - \beta_p)S_{p-1} + \beta_p k(\mathbf{x}_{i_p}, \mathbf{x})$
 6:     $\mathcal{I}_{p+1} \leftarrow \mathcal{I}_p \cup \{i_p\}$, $p \leftarrow p + 1$, set $\beta_p = 1/p$.
 7: **end for**

---

**Lemma A.2.** *The selected samples generated by Algorithm 1 satisfies*

$$\mathrm{MMD}_{k,\alpha}^2\left(\mathbf{X}_{\mathcal{I}_m}, \mathbf{X}_n\right) \leq M_\alpha^2 + B\frac{2 + \log m}{m + 1} \tag{9}$$

*where $B = 2K$, $0 \leq \max_{\mathbf{x} \in \mathcal{X}} k(\mathbf{x}, \mathbf{x}) \leq K$, $M_\alpha^2$ is defined by*

$$M_\alpha^2 := \min_{\mathbf{w}^\top \mathbf{1} = 1, \mathbf{w} \geq 0} \mathrm{MMD}_{k,\alpha,\mathbf{X}_n}^2(\mathbf{w})$$

*Proof.* Following the notations in Appendix A.1, let $\mathbf{p}_\alpha = \alpha\mathbf{p}$, we could straightly follow the proof for finite-sample-size error bound of kernel herding with predefined step sizes given by Pronzato (2021) to derive Lemma A.2, without any other innovation. $\square$

**Lemma A.3.** *Let $\mathcal{H}$ be a RKHS over $\mathcal{X}$ associated with positive definite kernel $k$, and $0 \leq \max_{\mathbf{x} \in \mathcal{X}} k(\mathbf{x}, \mathbf{x}) \leq K$. Let $\mathbf{X}_m = \{\mathbf{x}_i\}_{i=1}^m$, $\mathbf{Y}_n = \{\mathbf{y}_j\}_{j=1}^m$, $\mathbf{x}_i, \mathbf{y}_j \in \mathcal{X}$. Then for any $\alpha \leq 1$,*

$$|\mathrm{MMD}_{k,\alpha}(\mathbf{X}_m, \mathbf{Y}_n) - \mathrm{MMD}_k(\mathbf{X}_m, \mathbf{Y}_n)| \leq (1 - \alpha)\sqrt{K}$$

*Proof.*

$$|\mathrm{MMD}_{k,\alpha}(\mathbf{X}_m, \mathbf{Y}_n) - \mathrm{MMD}_k(\mathbf{X}_m, \mathbf{Y}_n)|$$

$$= \left| \sup_{\|f\|_\mathcal{H} \leq 1} \left( \frac{1}{m}\sum_{i=1}^m f(\mathbf{x}_i) - \frac{\alpha}{n}\sum_{j=1}^n f(\mathbf{y}_j) \right) - \sup_{\|f\|_\mathcal{H} \leq 1} \left( \frac{1}{m}\sum_{i=1}^m f(\mathbf{x}_i) - \frac{1}{n}\sum_{j=1}^n f(\mathbf{y}_j) \right) \right|$$

$$\leq \sup_{\|f\|_\mathcal{H} \leq 1} \left| \frac{1 - \alpha}{n}\sum_{i=1}^n f(y_i) \right| = \left( \frac{1 - \alpha}{n} \right) \sup_{\|f\|_\mathcal{H} \leq 1} \left| \sum_{i=1}^n f(y_i) \right|$$

$$= \left( \frac{1 - \alpha}{n} \right) \sup_{\|f\|_\mathcal{H} \leq 1} \left| \sum_{j=1}^n \langle f, k(\cdot, \mathbf{y}_j) \rangle_\mathcal{H} \right| \leq \left( \frac{1 - \alpha}{n} \right) \sup_{\|f\|_\mathcal{H} \leq 1} \sum_{j=1}^n |\langle f, k(\cdot, \mathbf{y}_j) \rangle_\mathcal{H}|$$

$$\leq \left( \frac{1 - \alpha}{n} \right) \sup_{\|f\|_\mathcal{H} \leq 1} \sum_{j=1}^n \|f\|_\mathcal{H}\|k(\cdot, \mathbf{y}_j)\|_\mathcal{H} \leq (1 - \alpha)\sqrt{K}$$

$\square$

**Lemma A.4** (Proposition 12.31, Wainwright (2019))**.** *Suppose that $\mathcal{H}_1$ and $\mathcal{H}_2$ are reproducing kernel Hilbert spaces of real-valued functions with domains $\mathcal{X}_1$ and $\mathcal{X}_2$, and equipped with kernels $k_1$ and $k_2$, respectively. Then the tensor product space $\mathcal{H} = \mathcal{H}_1 \otimes \mathcal{H}_2$ is an RKHS of real-valued functions with domain $\mathcal{X}_1 \times \mathcal{X}_2$, and with kernel function*

$$k\left((x_1, x_2), (x_1', x_2')\right) = k_1(x_1, x_1')\, k_2(x_2, x_2').$$

**Lemma A.5** (Theorem 5.7, Paulsen & Raghupathi (2016))**.** *Let $f \in \mathcal{H}_1$ and $g \in \mathcal{H}_2$, where $\mathcal{H}_1, \mathcal{H}_2$ be two RKHS containing real-valued functions on $\mathcal{X}$, which is associated with positive definite kernel $k_1, k_2$ and canonical feature map $\phi_1, \phi_2$, then for any $x \in \mathcal{X}$,*

$$f(x) + g(x) = \langle f, \phi_1(x) \rangle_{\mathcal{H}_1} + \langle g, \phi_2(x) \rangle_{\mathcal{H}_2} = \langle f + g, (\phi_1 + \phi_2)(x) \rangle_{\mathcal{H}_1 + \mathcal{H}_2}$$

*where*

$$\mathcal{H}_1 + \mathcal{H}_2 = \{f_1 + f_2 | f_i \in \mathcal{H}_i\}$$

*and $\phi_1 + \phi_2$ is the canonical feature map of $\mathcal{H}_1 + \mathcal{H}_2$. Furthermore,*

$$\|f + g\|_{\mathcal{H}_1 + \mathcal{H}_2}^2 \leq \|f\|_{\mathcal{H}_1}^2 + \|g\|_{\mathcal{H}_2}^2$$

**Lemma A.6.** *For any unlabeled dataset $\mathbf{X}_n \subset \mathcal{X}$ and selected samples $\mathbf{X}_{\mathcal{I}_m}$,*

$$\mathrm{MMD}_{k,\alpha}^2(\mathbf{X}_n, \mathbf{X}_n) = (1 - \alpha)^2 \overline{K}, \mathrm{MMD}_{k,\alpha}^2(\mathbf{X}_{\mathcal{I}_m}, \mathbf{X}_n) \leq (1 + \alpha^2) K$$

*where $\overline{K} = \sum_{i=1}^n \sum_{j=1}^n k(\mathbf{x}_i, \mathbf{x}_j)/n^2$, $K = \max_{\mathbf{x} \in \mathcal{X}} k(\mathbf{x}, \mathbf{x})$.*

Lemma A.6 is immediately proved by the definition of $\alpha$-MMD.

### A.3 PROOF OF THEOREMS

*Proof for Theorem 3.1.* Firstly, let us denote that $\mathcal{H}_4 = \mathcal{H}_1 \otimes \mathcal{H}_1 + \mathcal{H}_1 \otimes \mathcal{H}_2 + \mathcal{H}_3$, with kernel $k_4 = k_1^2 + k_1 k_2 + k_3$ and canonical feature map $\phi_4 = \phi_1 \otimes \phi_1 + \phi_1 \otimes \phi_2 + \phi_3$.

Under the assumptions in Theorem 3.1, according to Theorem 4 in Song et al. (2009), we have for any $\mathbf{x} \in \mathcal{X}$,

$$h(\mathbf{x}) = \langle h, \phi_1(\mathbf{x}) \rangle_{\mathcal{H}_1}, \mathbb{E}[Y|\mathbf{x}] = \langle \mathbb{E}[Y|X], \phi_2(\mathbf{x}) \rangle_{\mathcal{H}_2},$$
$$\mathrm{Var}(Y|\mathbf{x}) = \langle \mathrm{Var}(Y|X), \phi_3(\mathbf{x}) \rangle_{\mathcal{H}_3}$$

where $\phi_1, \phi_2, \phi_3$ are canonical feature maps in $\mathcal{H}_1, \mathcal{H}_2, \mathcal{H}_3$. Denote that $m = \mathbb{E}[Y|X]$ and $s = \mathrm{Var}(Y|X)$. Now by definition,

$$R(h) = \mathbb{E}\left[\ell(h(\mathbf{x}), y)\right] = \int_{\mathcal{X}} \int_{\mathcal{Y}} \ell(h(\mathbf{x}), y) p(y|\mathbf{x}) p(\mathbf{x}) d\mathbf{x} dy = \int_{\mathcal{X}} f(\mathbf{x}) p(\mathbf{x}) d\mathbf{x}$$

where

$$
\begin{aligned}
f(x) &= \int_{\mathcal{Y}} (y - h(\mathbf{x}))^2 p(y|\mathbf{x}) dy \\
&= \mathrm{Var}(Y|\mathbf{x}) + 2h(\mathbf{x})\mathbb{E}[Y|\mathbf{x}] + h^2(\mathbf{x}) \\
&= \langle s, \phi_3(\mathbf{x}) \rangle_{\mathcal{H}_3} + 2 \langle h, \phi_1(\mathbf{x}) \rangle_{\mathcal{H}_1} \langle m, \phi_2(\mathbf{x}) \rangle_{\mathcal{H}_2} + \langle h, \phi_1(\mathbf{x}) \rangle_{\mathcal{H}_1} \langle h, \phi_1(\mathbf{x}) \rangle_{\mathcal{H}_1} \\
&= \langle s, \phi_3(\mathbf{x}) \rangle_{\mathcal{H}_3} + \langle 2h \otimes m, (\phi_1 \otimes \phi_2)(\mathbf{x}) \rangle_{\mathcal{H}_1 \otimes \mathcal{H}_2} + \langle h \otimes h, (\phi_1 \otimes \phi_1)(\mathbf{x}) \rangle_{\mathcal{H}_1 \otimes \mathcal{H}_1} \\
&= \langle s + 2h \otimes m + h \otimes h, \phi_5(x) \rangle_{\mathcal{H}_4}
\end{aligned}
$$

where the fourth equality holds by Lemma A.4 and the last equality holds by Lemma A.5, then $f \in \mathcal{H}_4$, and

$$
\begin{aligned}
\|f\|_{\mathcal{H}_4} &= \|s + 2h \otimes m + h \otimes h\|_{\mathcal{H}_4} \\
&\leq \|s\|_{\mathcal{H}_4} + \|2h \otimes m\|_{\mathcal{H}_4} + \|h \otimes h\|_{\mathcal{H}_4} \\
&\leq \|s\|_{\mathcal{H}_3} + 2\|m\|_{\mathcal{H}_2}\|h\|_{\mathcal{H}_1} + \|h \otimes h\|_{\mathcal{H}_1 \otimes \mathcal{H}_1} \\
&= \|s\|_{\mathcal{H}_3} + 2\|m\|_{\mathcal{H}_2}\|h\|_{\mathcal{H}_1} + \|h\|_{\mathcal{H}_1}^2 \\
&\leq K_h^2 + 2K_h K_m + K_s
\end{aligned}
$$

where the second inequality holds by Lemma A.5. Therefore, let $\beta = 1/(K_h^2 + 2K_h K_m + K_s)$ we have $\|\beta f\|_{\mathcal{H}_4} = \beta \|f\|_{\mathcal{H}_4} \leq 1$. Then

$$
\begin{aligned}
& \left| \widehat{R}_T(h) - \widehat{R}_S(h) \right| \\
&= \left| \int_{\mathcal{X}} f(\mathbf{x}) dP_T(\mathbf{x}) - \int_{\mathcal{X}} f(\mathbf{x}) dP_S(\mathbf{x}) \right| \\
&= (K_h^2 + 2K_h K_m + K_s) \left| \int_{\mathcal{X}} \beta f(\mathbf{x}) dP_T(\mathbf{x}) - \int_{\mathcal{X}} \beta f(\mathbf{x}) dP_S(\mathbf{x}) \right| \\
&\leq (K_h^2 + 2K_h K_m + K_s) \sup_{\|f\|_{\mathcal{H}_4} \leq 1} \left| \int_{\mathcal{X}} f(\mathbf{x}) dP_T(\mathbf{x}) - \int_{\mathcal{X}} f(\mathbf{x}) dP_S(\mathbf{x}) \right| \\
&= (K_h^2 + 2K_h K_m + K_s) \mathrm{MMD}_{k_4}(\mathbf{X}_S, \mathbf{X}_T)
\end{aligned}
$$

where $P_T$ denotes the empirical distribution constructed by $\mathbf{X}_T$, so does $P_S$. Recall Lemma A.3, we have Theorem 3.1. $\qquad\square$

*Proof for Theorem 3.2.* Following the notations in Appendix A.1, we further define

$$\mathbf{w}_* = \mathbf{1}/n, C_\alpha^2 = \mathrm{MMD}_{k,\alpha,\mathbf{X}_n}^2(\mathbf{w}_*) = (1-\alpha)^2\overline{K} \tag{10}$$

$$\widehat{\mathbf{w}} = \arg\min_{\mathbf{1}^\top\mathbf{w}=1} \mathrm{MMD}_{k,\alpha,\mathbf{X}_n}^2(\mathbf{w}) = \alpha\left(\mathbf{K}^{-1} - \frac{\mathbf{K}^{-1}\mathbf{1}\mathbf{1}^\top\mathbf{K}^{-1}}{\mathbf{1}^\top\mathbf{K}^{-1}\mathbf{1}}\right)\mathbf{p} + \frac{\mathbf{K}^{-1}\mathbf{1}}{\mathbf{1}^\top\mathbf{K}^{-1}\mathbf{1}}$$

Let $\mathbf{p}_\alpha = \alpha\mathbf{p}$, we have $(\mathbf{p}_\alpha - \mathbf{K}\widehat{\mathbf{w}}) \propto \mathbf{1}$. Define

$$\Delta_\alpha(\mathbf{w}) := \mathrm{MMD}_{k,\alpha,\mathbf{X}_n}^2(\mathbf{w}) - C_\alpha^2 = \widehat{g}(\mathbf{w}) - \widehat{g}(\mathbf{w}_*)$$

where $\widehat{g}(\mathbf{w}) = (\mathbf{w} - \widehat{\mathbf{w}})^\top\mathbf{K}(\mathbf{w} - \widehat{\mathbf{w}})$. The related details for proving the equality are omitted, since they are completely given by the proof of alternative expression of MMD in Pronzato (2021). By the convexity of $\widehat{g}(\cdot)$, for $j = \arg\min_{i\in[n]\setminus\mathcal{I}_p} f_{\mathcal{I}_p}(\mathbf{x}_i)$,

$$\widehat{g}\left(\mathbf{w}_*\right) \geq \widehat{g}\left(\mathbf{w}_p\right) + 2\left(\mathbf{w}_* - \mathbf{w}_p\right)^\top\mathbf{K}\left(\mathbf{w}_p - \widehat{\mathbf{w}}\right) \geq \widehat{g}\left(\mathbf{w}_p\right) + 2\min_{j\in[n]\setminus\mathcal{I}_p}\left(\mathbf{e}_j - \mathbf{w}_p\right)^\top\mathbf{K}\left(\mathbf{w}_p - \widehat{\mathbf{w}}\right)$$

where the second inequality holds with the assumption in Theorem 3.2

$$\begin{aligned}
\left(\mathbf{w}_* - \mathbf{e}_j\right)^\top\mathbf{K}\left(\mathbf{w}_p - \widehat{\mathbf{w}}\right) &= \left(\mathbf{w}_* - \mathbf{e}_j\right)^\top\left(\mathbf{K}\mathbf{w}_p - \mathbf{p}_\alpha\right)\\
&= \frac{\sum_{i=1}^n f_{\mathcal{I}_p}(\mathbf{x}_i)}{n} - f_{\mathcal{I}_p}(\mathbf{x}_{j_{p+1}}) \geq \frac{\sum_{i=1}^n f_{\mathcal{I}_p}(\mathbf{x}_i)}{n} - f_{\mathcal{I}_p}(\mathbf{x}_j) \geq 0
\end{aligned}$$

therefore, we have for $B = 2K$,

$$\begin{aligned}
&\Delta_\alpha(\mathbf{w}_{p+1})\\
=&\widehat{g}\left(\mathbf{w}_p\right) - \widehat{g}\left(\mathbf{w}_*\right) + \frac{2}{p+1}\left(\mathbf{e}_j - \mathbf{w}_p\right)^\top\mathbf{K}\left(\mathbf{w}_p - \widehat{\mathbf{w}}\right) + \frac{1}{(p+1)^2}\left(\mathbf{e}_j - \mathbf{w}_p\right)^\top\mathbf{K}\left(\mathbf{e}_j - \mathbf{w}_p\right)\\
=&\frac{p}{p+1}\left(\widehat{g}\left(\mathbf{w}_p\right) - \widehat{g}\left(\mathbf{w}_*\right)\right) + \frac{1}{(p+1)^2}B = \frac{p}{p+1}\Delta_\alpha(\mathbf{w}_p) + \frac{1}{(p+1)^2}B
\end{aligned}$$
$$\tag{11}$$

where $\mathbf{w}_{p+1} = p\mathbf{w}_p/(p+1) + \mathbf{e}_j/(p+1)$, and obviously $B$ upper bounds $\left(\mathbf{e}_j - \mathbf{w}_p\right)^\top\mathbf{K}\left(\mathbf{e}_j - \mathbf{w}_p\right)$. Since $\alpha \leq 1$, it holds from Lemma A.6 that

$$\Delta_\alpha(\mathbf{w}_1) \leq \mathrm{MMD}_{k,\alpha,\mathbf{X}_n}^2(\mathbf{w}_1) \leq (1+\alpha^2)K \leq B$$

therefore by Lemma A.1, we have

$$\mathrm{MMD}_{k,\alpha}^2(\mathbf{X}_{\mathcal{I}_m}, \mathbf{X}_n) = \mathrm{MMD}_{k,\alpha,\mathbf{X}_n}^2(\mathbf{w}_p) \leq C_\alpha^2 + B\frac{2+\log m}{m+1}$$

$\qquad\square$

## A.4 NUMERICAL STUDIES ON GKHR

Firstly, we define four distributions on $\mathbb{R}^2$:

1. Gaussian mixture model 1 which consists of four Gaussian distributions $G_1, G_2, G_3, G_4$ with mixture weights $[0.95, 0.01, 0.02, 0.02]$,

2. Gaussian mixture model 2 which consists of four Gaussian distributions $G_1, G_2, G_3, G_4$ with mixture weights $[0.3, 0.2, 0.15, 0.35]$,

3. Uniform distribution 1 which consists of a uniform distribution defined in a circle with radius 0.5, and a uniform distribution defined in a annulus with inner radius 4 and outer radius 6,

4. Uniform distribution 2 defined on $[-10, 10]^2$.

where

$$G_1 = \mathcal{N}\left(\begin{bmatrix} 1 \\ 2 \end{bmatrix}, \begin{bmatrix} 2 & 0 \\ 0 & 5 \end{bmatrix}\right), G_2 = \mathcal{N}\left(\begin{bmatrix} -3 \\ -5 \end{bmatrix}, \begin{bmatrix} 1 & 0 \\ 0 & 2 \end{bmatrix}\right)$$

$$G_3 = \mathcal{N}\left(\begin{bmatrix} -5 \\ 4 \end{bmatrix}, \begin{bmatrix} 8 & 0 \\ 0 & 6 \end{bmatrix}\right), G_4 = \mathcal{N}\left(\begin{bmatrix} 15 \\ 10 \end{bmatrix}, \begin{bmatrix} 4 & 0 \\ 0 & 9 \end{bmatrix}\right)$$

To consistently evaluate the performance gap between GKHR and GKH at the same order of magnitude, we propose the following criterion

$$D = \frac{D_1 - D_2}{D_1 + D_2}$$

where $D_1 = \text{MMD}^2_{k,\alpha}(\mathbf{X}^{(1)}_{\mathcal{I}_m}, \mathbf{X}_n), D_2 = \text{MMD}^2_{k,\alpha}(\mathbf{X}^{(2)}_{\mathcal{I}_m}, \mathbf{X}_n)$, $\mathbf{X}^{(1)}_{\mathcal{I}_m}$ is the selected samples from GKHR and $\mathbf{X}^{(2)}_{\mathcal{I}_m}$ is the selected samples from GKH. Positive value of $D$ implies that GKH outperforms GKHR, and negative values of $D$ implies that GKHR outperforms GKH. Large absolute value of $D$ shows large performance gap.

The experiments are conducted as follows. We generate 1000,3000,10000,30000 random samples from the four distributions separately, then use GKHR and GKH for sample selection in the low-budget setting, i.e., $m/n \leq 0.2$. We report the results over ten independent runs in Figure 3, which shows that although the performance gap tends to grow as $m$ grows, when $m$ is relatively small, the performance of GKHR is similar to that of GKH.

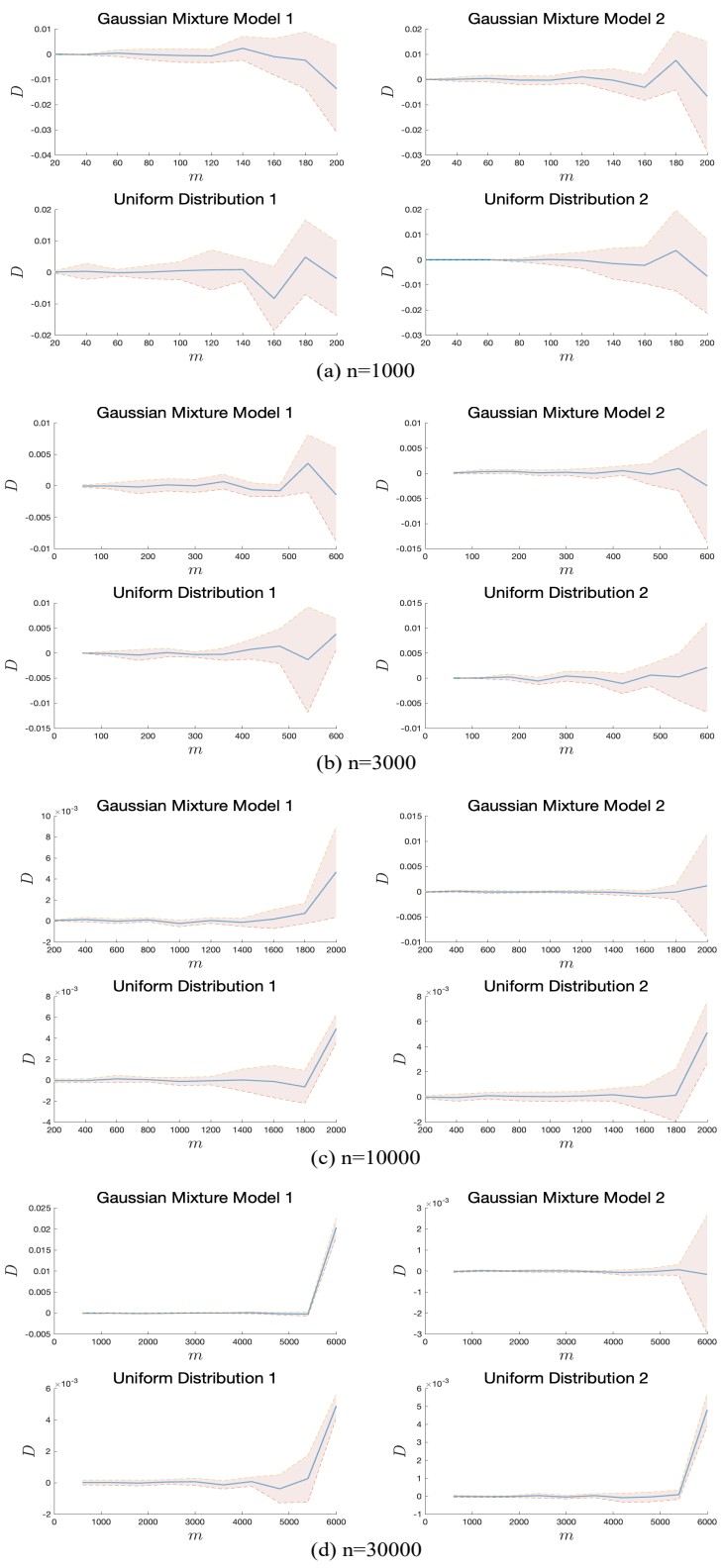

Figure 3: The performance comparison between GKHR and GKH with different $m, n$ over ten independent runs. The blue line is the mean value of $D$, the red dotted line over (under) the blue line is the mean value of $D$ plus (minus) its standard deviation, and the pink area is the area between the upper and lower red dotted lines.

