# OpenReview forum: "Unleashing the Power of Annotation: Enhancing Semi-Supervised Learning through Unsupervised Sample Selection"
_ICLR.cc/2024/Conference — Submitted to ICLR 2024_

### Official Review · Reviewer_VKdd · 2023-10-31

**Soundness:** 2 fair
**Presentation:** 2 fair
**Contribution:** 1 poor
**Rating:** 3
**Confidence:** 3

**Summary:**

The paper at hand proposes a sampling approach for selecting a smaller but still representative and diverse set from a large dataset. This is applied to the task of semi-supervised learning where the subset selection is applied to unlabeled data which is to be labeled later on.

**Strengths:**

+ relevant problem

**Weaknesses:**

- limited comparison to other sampling methods (only random sampling)
- marginal improvement over random sampling (Tab. 1 overlapping confidence intervals)
- not clearly and convincedly presented advantages of the method (e.g., Fig. 2, it's hard to see the claimed benefits)

Honestly speaking ,I'm not sure that the chosen application is the right one. I would recommend focusing on the selection step to approximate the distribution (estimated from the large dataset) with few examples. Compare the proposed approach to related work, discuss pros and cons. Finally sketch various applications (including SSL) briefly which will benefit.

**Questions:**

How does the approach scope with low and high dimensions?
How does the approach scope with different distributions (overlapping, clusters well separated, etc.)?
What are limitations?

---

> ### Author Response · Authors · 2023-11-22
> **Responses To Weaknesses**
>
> Thank you very much for your comments. We hope our responses can address all your concerns.
>
> **1. Limited comparison to other sampling methods (only random sampling).**
>
> We are currently in the process of replicating some other methods, and we will provide a response with the results once this task is completed.
>
> **2. Marginal improvement over random sampling (Tab. 1 overlapping confidence intervals).**
>
> Current semi-supervised algorithms have already achieved relatively superior performance, making further enhancements a challenging and complex endeavor. Our methodology demonstrates limited improvements when the annotation budget is substantial. However, it shows considerable advancements in scenarios with constrained budgets. As shown in Table 1 of the manuscript, when employing UPA, Flexmatch achieves an accuracy improvement of 4.4% on the CIFAR-10 dataset with an annotation budget of 40, a 2.5% increase on the CIFAR-100 dataset with a budget of 400, and a 3.6% rise on the SVHN with a budget of 250. These findings align with intuitive conjecture: as the annotation budget increases, regardless of the sampling method employed, there is a greater likelihood of acquiring a more comprehensive set of samples. Consequently, the disparity between different sampling methods diminishes. Therefore, the superiority of a sampling method becomes more pronounced with a smaller annotation budget.
>
> **3. Not clearly and convincedly presented advantages of the method (e.g., Fig. 2, it's hard to see the claimed benefits).**
>
> We have introduced a parameter $\alpha$ to encourage diversity among the selected samples, ensuring that they are as dissimilar as possible. This approach prevents the over-concentration of selected samples and thereby avoids label redundancy. Figure 2 illustrates how our method achieves better space-filling property in sampling within different categories. In contrast to the other two methods, our approach does not result in sample clustering. We will given a more specified explanation in the revised version.
>
> **4. Focus on the selection step to approximate the distribution (estimated from the large dataset) with few examples. Compare the proposed approach to related work, discuss pros and cons. Finally sketch various applications (including SSL) briefly which will benefit.**
>
> The process of sample selection in AL is inherently coupled with model training. Thus, the selection procedure iterates in tandem with the model training, where a subset of samples is chosen in each iteration for labeling and training until the budget is exhausted. However, our approach entails a one-time selection of all samples for labeling. This results in our sample selection algorithm being inapplicable to the AL paradigm, thereby posing challenges in direct comparison with other AL methods.
>
> In order to provide a more intuitive comparison between our sampling method and AL sampling methods in Related Work, while considering that AL sampling methods only utilize labeled data for supervised training, we devised a new comparative experiment under the same labeling budget constraint. First, we employed our method to select samples for annotation, followed by using these annotated samples for supervised classifier training. This step was undertaken to demonstrate the performance contrast between different sampling methods in supervised learning. The table below presents the experimental results.
>
> | Methods | Budget | Acc (%) |
> | --- | --- | --- |
> | *Active Learning* |  |  |
> | VAAL | 7500 | 86.8 |
> | UncertainGCN | 7500 | 86.8 |
> | CoreGCN | 7500 | 86.5 |
> | MCDAL | 7500 | 87.2 |
> | *Supervised Learning* |  |  |
> | Ours | 7500 | 86.6 |
> ||||
>
> It is worth noting that within AL framework, sample selection is tightly coupled with model training, where model training is dynamic, and subsequent sample selection becomes more precise as performance improves. This is an iterative process that continues until the labeling budget is exhausted. In contrast to our sampling method, all samples that require annotation are determined before training (*i.e.,* in the absence of knowledge about the training data). The challenge with this approach is that all samples must be selected initially, and there is no flexibility to adjust sample selection based on intermediate training results. Although theoretically, this may lead to lower performance compared to dynamic sample selection in AL methods, experimental results demonstrate that even under a static selection framework, our method can achieve performance similar to AL. This indirectly reflects the advantages of our method.

---

> ### Author Response · Authors · 2023-11-22
> **Responses To Questions**
>
> **5. How does the approach scope with low and high dimensions? How does the approach scope with different distributions (overlapping, clusters well separated, etc.)? What are limitations?**
>
> a) Using MMD to measure the representativeness is not influenced by the dimension of feature space theoretically, since MMD is actually the distance between kernel mean embeddings of full data and selected samples in a RKHS. Similarly, $\alpha$-MMD also enjoys such property. Therefore, the approach is not sensitive to the dimension of feature space.
>
> b) The "optimal choice" is actually a recommended choice, and derived only from theoretical aspect. However, in practice, this choice seems to be effective. For well separated clusters, the algorithm is encouraged to select more data points lying on the boundaries of the clusters, which are more informative than the data points that lie aside the center of clusters. For overlapped distributions, since in the assumptions of SSL, few points are close to each other but in different classes, diversification is still a good idea for selecting informative samples. Consider that diversification prefer to select boundary points, low-density-region points and points from small clusters, the representativeness ensures that the proportion of selected points from each cluster is not too different from the proportion of full data. Nevertheless, if the cluster are sufficiently far from each other, we could encourage more diversity via decreasing the value of $\alpha$ so that more boundary points are selected.
>
> c) A limitation of our method is that there is no certain criteria for choosing $\alpha$ that consider the distribution of full data. In b) we find that for overlapped distributions and clusters-well-separated distributions, choosing $\alpha$ could possibly follow different strategies. In other words, there could be a better choice (such as some distribution-dependent ones) for $\alpha$ than that we recommend.

---

> > ### Comment · Area_Chair_zqak · 2023-11-23
> >
> > Dear Reviewer VKdd,
> >
> > The authors have responded to your reviews. Can you please take a look at their response to see if your concerns were addressed and whether you would like to update your score?
> >
> > Best regards,
> >  - Your AC.

---

### Official Review · Reviewer_Xj3X · 2023-11-01

**Soundness:** 1 poor
**Presentation:** 1 poor
**Contribution:** 2 fair
**Rating:** 3
**Confidence:** 5

**Summary:**

The paper suggests a maximum mean discrepancy approach to select data for annotation. The approach, referred to as UPA, attempts to capture representative and diverse set of point to improve on active learning. Some theoretical results are presented based on a similar approach to Coarset (Sener et al.) to reduce the risk on the labelled set by selecting a coareset.
Empirical validation is presented for a few benchmarks and comparison with a few  active learning approaches and SSAL.

**Strengths:**

1.	An approach the optimizes both diversity and representativeness has advantage
2.	Some theoretical results support the advantage of the method.
3.	Empirical results show some advantage over Random baseline and for cifar 10 clear advantage to using flexMatch with UPA

**Weaknesses:**

1.	The soundness of the paper is poor:

a.	mostly the empirical validation is lacking additional datasets to support the advantage of the method

b.	improvement over random baseline is too modest in my view, 0.5 precent improvement when the STD is at 0.5 is not convincing

c.	we are missing a simple ablation study: if you show results of flexMatch+UPA, you have to show also results of just flexMatch, otherwise it is not clear if the advantage is due to FlexMatch or the addition of UPA.

d.	3 independent runs is not enough

e.	Why are you selecting the particular m values for each data set? why is it different for each data set? I would much prefer seeing a graph over a range of values.

f.	Not enough method are used in the comparison, what about BADGE by Ash et al -  it is a classical diversity and uncertainty approach that could also be compared to show that representativeness and diversity is better (if it is indeed so…)

2.	The requirement that m/n <=0.2 is quite weak and unrealistic, sometimes 0.2 can be a huge data set to annotate! How does that work with Active learning setting in which the budget is limited.
3.	Clarity is also poor:

a.	There is no pseudo code describing the method, or at least a set of steps

b.	What is the input space over which the method is used? Is it the actual data? For coreset is it the penultimate layer representation, what are you using?

c.	Remark 1 is rather confusing, if you present Thm 3.1 and then claim that thm 3.1 doesn’t always work

d.	Figure 1 can not be deciphered with out a basic explanation\legend for the colors

e.	Equation 6: is f_I_p(x_i) why is it defined only for x?

f.	In equation (8) what is K in B=2K?

4.	The computational complexity of O(mn) should be better explained? Wouldn’t the kernel construction cost more?

5.	Overall Im not convinced at all that the approach of diversification and representativeness is an optimal one. There are other important trade offs in active learning such as exploration-exploitation, which can outperform this UPA approach, in my view.

**Questions:**

1.	Please see the questions above in the ‘weaknesses’
2.	Please explain what does the flexMatch method do , and why you chose it for your UPA approach.
3.	Why do you think the diversification and representativeness is the best one for AL?

---

> ### Author Response · Authors · 2023-11-22
> **Responses**
>
> Thank you very much for your comments. We hope our responses can address all your concerns.
>
> **1. Mostly the empirical validation is lacking additional datasets to support the advantage of the method.**
>
> Your point regarding the importance of additional datasets for validating the strengths of our method is very pertinent. In this paper, due to time and resource constraints, we selected CIFAR-10, CIFAR-100, and SVHN as our primary datasets for experiments. These datasets were chosen because of their wide acceptance and representativeness in the SSL domain. Our method demonstrated significant improvements on these datasets, especially in scenarios with a limited labeling budget. However, we acknowledge that applying our method to a broader and more diverse set of datasets is necessary for a more comprehensive validation of its strengths. To this end, we have initiated preliminary experiments on the STL-10 and ImageNet datasets, which is expected to take some time. We will provide a detailed response with the results upon the completion of this task.
>
> **2. Improvement over random baseline is too modest in my view, 0.5 precent improvement when the STD is at 0.5 is not convincing.**
>
> Current semi-supervised algorithms have already achieved relatively superior performance, making further enhancements a challenging and complex endeavor. Our methodology demonstrates limited improvements when the annotation budget is substantial. However, it shows considerable advancements in scenarios with constrained budgets. As shown in Table 1 of the manuscript, when employing UPA, Flexmatch achieves an accuracy improvement of 4.4% on the CIFAR-10 dataset with an annotation budget of 40, a 2.5% increase on the CIFAR-100 dataset with a budget of 400, and a 3.6% rise on the SVHN with a budget of 250. These findings align with intuitive conjecture: as the annotation budget increases, regardless of the sampling method employed, there is a greater likelihood of acquiring a more comprehensive set of samples. Consequently, the disparity between different sampling methods diminishes. Therefore, the superiority of a sampling method becomes more pronounced with a smaller annotation budget.
>
> **3. We are missing a simple ablation study: if you show results of flexMatch+UPA, you have to show also results of just flexMatch, otherwise it is not clear if the advantage is due to FlexMatch or the addition of UPA.**
>
> In the paradigm of Semi-Supervised Learning (SSL), a large amount of unlabeled data is initially provided, from which a subset is selected for labeling. The SSL algorithms, such as FlexMatch/FreeMatch used in our study, then leverage both the labeled and remaining unlabeled data for training. Therefore, any SSL algorithm cannot function independently without a sample selection method. The most widely used sample selection strategies currently are random sampling and stratified sampling. Stratified sampling necessitates prior knowledge of each sample's category for performing random sampling within each category, which is impractical in real-world scenarios. Consequently, random sampling serves as a feasible baseline method for comparison with our approach.
>
> **4. 3 independent runs is not enough.**
>
> Thank you for your comment. We will increase the number of experiments conducted to reduce the randomness in the experimental results.
>
> **5. Why are you selecting the particular m values for each data set? why is it different for each data set? I would much prefer seeing a graph over a range of values.**
>
> In the domain of SSL, most of the popular algorithms adopt such experimental settings, as referenced in [1]-[3].
>
> [1] https://proceedings.neurips.cc/paper/2020/file/06964dce9addb1c5cb5d6e3d9838f733-Paper.pdf
> [2] https://proceedings.neurips.cc/paper/2021/file/995693c15f439e3d189b06e89d145dd5-Paper.pdf
> [3] https://openreview.net/pdf?id=PDrUPTXJI_A
>
>
> **6. Not enough method are used in the comparison, what about BADGE by Ash et al - it is a classical diversity and uncertainty approach that could also be compared to show that representativeness and diversity is better (if it is indeed so…).**
>
> Thanks for your suggestions, we are adding comparative experiments with BADGE.
>
> **7. The requirement that m/n <=0.2 is quite weak and unrealistic, sometimes 0.2 can be a huge data set to annotate! How does that work with Active learning setting in which the budget is limited.**
>
> We conducted comparative experiments under different labeling budgets, where m/n=0.2 represents the scenario with the maximum labeling budget. In most comparative experiments, the value of m/n is much less than 0.2. For instance, on the CIFAR-10 dataset with a budget of 40, the ratio is m/n=40/50000=0.0008; similarly, on the SVHN dataset with a budget of 250, it is m/n=250/73257=0.0034.
>
> **8. There is no pseudo code describing the method, or at least a set of steps.**
>
> Due to space limitations, we display the pseudocode in the appendix.

---

> ### Author Response · Authors · 2023-11-22
> **Responses**
>
> **9. What is the input space over which the method is used? Is it the actual data? For coreset is it the penultimate layer representation, what are you using?**
>
> The input of the sampling algorithm is the output of the image feature extractor. The specific process is as follows. First, we leverage the pre-trained image feature extraction capabilities of CLIP, which is a vision transformer architecture, to extract features. Subsequently, the [CLS] token features produced by the model’s final output are employed for sample selection.
>
> **10. Remark 1 is rather confusing, if you present Thm 3.1 and then claim that thm 3.1 doesn’t always work.**
>
> Thm 3.1. claims that if the model (or hypothesis class) belongs to a certain RKHS, then the expected risk of the trained model (or hypothesis) is controlled. If not, then the theory is not valid. Since the methodology is designed to be model-free, we think it would be better if claim that Thm 3.1 does not work for every model.
>
> **11. Figure 1 can not be deciphered with out a basic explanation\legend for the colors.**
>
> Thank you for your comment, we will supplement the legend of Figure 1 in the revised version.
>
> **12. Equation 6: is f_I_p(x_i) why is it defined only for x?**
>
> This is definitely a typo. We are really sorry for that. Substitute $x_i$ by $x$ would be right. We will correct that in the revised version. Thank you for your carefulness.
>
> **13. In equation (8) what is K in B=2K?**
>
> K is defined in Thm 3.2. It is the upper bound of k(x,x) over $\mathcal{X}$.
>
> **14. The computational complexity of O(mn) should be better explained? Wouldn’t the kernel construction cost more?**
>
> The result of computational complexity almost directly follows from the result of https://arxiv.org/pdf/2101.07564.pdf. Please refer to Section 3.1. However, after numerical experiments, we find that the kernel construction indeed cost more time in high-dimensional cases. Therefore, we may revise the computation complexity by O(mnd) where d is the dimension of feature space. Thanks.
>
> **15. Overall Im not convinced at all that the approach of diversification and representativeness is an optimal one. There are other important trade offs in active learning such as exploration-exploitation, which can outperform this UPA approach, in my view.**
>
> Diversity as a criteria of selecting samples in semi-supervised learning is theoretically studied by https://doi.org/10.1080/01621459.2023.2282644, and the simulation results in this paper shows that diversification indeed works. Representativeness is a straightforward idea to measure the sampling quality. Thus our criteria for the unsupervised sample selection task could be proper. As for optimality, we think that there does not exists optimal strategy for such unsupervised task. The exploration-exploitation framework in active learning is indeed effective and ingenious, but our unsupervised sample selection task does not require feedbacks from the model, which makes some active learning framework improper for semi-supervised learning.
>
> **16. Please explain what does the flexMatch method do , and why you chose it for your UPA approach.**
>
> FlexMatch and FreeMatch are state-of-the-art SSL algorithms that already achieve commendable results when implemented with baseline sampling methods, such as random sampling. Enhancing their performance further is a challenging task. Our method, however, manages to improve their performance further. Therefore, choosing them serves to more convincingly demonstrate the effectiveness of our approach.
>
> **17. Why do you think the diversification and representativeness is the best one for AL?**
>
> In this paper we mainly focus on the methodology of unsupervised sample selection and its application in semi-supervised learning tasks, nevertheless, we did not discuss its application in AL tasks. Maybe diversification and representativeness could be a good choice for AL, as a extension of core-set approach (only representativeness considered) proposed in https://arxiv.org/abs/1708.00489, which is a well-recognise work in the community.

---

### Official Review · Reviewer_VfL6 · 2023-11-01

**Soundness:** 3 good
**Presentation:** 3 good
**Contribution:** 2 fair
**Rating:** 5
**Confidence:** 3

**Summary:**

The paper presents the method of selecting instances for annotation, based on the MMD (Max Mean Discrepancy) principle. According to the principle, the method aims to minimize the distance between the full unlabeled dataset, and the sampled instances. An additional parameter $\alpha$ is introduced to tradeoff sample representativeness with diversity. The Kernel Herding algorithm is used to iteratively find the target set of points.

**Strengths:**

Selecting an optimal subset for annotation is an important problem in scenarios where labels are costly to get. The main contribution of the paper is the demonstration that minimizing MMD for such scenarios helps improve underlying classification accuracy. While there is a clear parallel with the coreset constriction idea, the paper gives a theoretical result which relates the two approaches. The experimental section provides a set of comparisons with the baselines which demonstrate advantages of using MMD.

**Weaknesses:**

Introducing the parameter $\alpha$ doesn’t seem to have enough theoretical or experimental grounding. From the theoretical standpoint, the trivial case $\alpha=1$ makes all bounds tighter than for any other alpha. From the experimental part, the only ablation analysis of $\alpha$ is given in table 3, it doesn’t fully convince that using any other value except $\alpha=1$ is any better. Specifically, the results for $\alpha=1$ and "optimal" $\alpha=1-1/\sqrt{m}$ are quite close, and measured only on 3 datapoints for one dataset. More experimental grounding for motivating the choice $\alpha=1-1/\sqrt{m}$ over $\alpha=1$ would help.

Without taking $\alpha$ into consideration, there is not much novelty in the introduced methods. Overall, the paper provides good justification of using MMD for sample selection.

**Questions:**

In the experimental section, it would be great to have the results of a supervised classifier (not SSL) trained on the selected set of instances, and see at what subset size the accuracy would match the accuracy of a classifier trained on the whole training set.

---

> ### Author Response · Authors · 2023-11-16
> **Responses To Weaknesses**
>
> Thank you very much for your comments. We hope our responses can address all your concerns.
>
>
> **1. Introducing the parameter $\alpha$ doesn’t seem to have enough theoretical or experimental grounding. From the theoretical standpoint, the trivial case $\alpha=1$ makes all bounds tighter than for any other alpha. From the experimental part, the only ablation analysis of $\alpha$ is given in table 3, it doesn’t fully convince that using any other value except $\alpha=1$ is any better.**
>
> We will explain the necessity of introducing $\alpha$ from two aspects.
>
> **1.1 From a theoretical perspective**
>
> Theoretically $\alpha=1$ makes all bounds mentioned in this paper tighter, but this degenerates $\alpha$-MMD into MMD, which lead to the lack of diversity of samples selected by GKHR algorithm. Hence, even though setting $\alpha=1$ benefits the tightness of bounds, it violates the main goal of our idea. Actually, we prefer to regrad $\alpha$ as a predefined parameter which only depends on our preference for the diversity of selected samples. Nevertheless, the analysis on the bound helps us to derive a optimal choice of $\alpha$, which makes the selected samples as diverse as possible under the premise of their representativeness. The word "optimal" could be exaggerate here, but we still emphasize that it is recommended. We think substitute "optimal choice" by "recommended choice" to describe the setting $\alpha=1-1/\sqrt{m}$ in the revised version could be more proper.
>
> **1.2 From an experimental perspective**
>
> In fact, we have also conducted experiments with various values of $\alpha$ on CIFAR-10. However, due to the constraints of the initial submission deadline, these results were not included in the manuscript submitted. The experimental results on CIFAR-10 are reported below, and the setting of $\alpha=1-1/\sqrt{m}$ consistently achieved the highest Top-1 classification accuracy under three different annotation budget scenarios (*i.e.* $40$, $250$ and $4000$).
>
> | $\alpha$ |  40  |   250  |  4000  |
> | ---  | --- | --- | --- |
> | 0.8  | 94.2 $\pm$ 0.4       | 94.8 $\pm$ 0.4        | 94.9 $\pm$ 0.2 |
> | 0.9  | 94.1 $\pm$ 0.1       | 95.1 $\pm$ 0.2        | 94.9 $\pm$ 0.3 |
> | 0.95 | 94.0 $\pm$ 0.4       | 94.9 $\pm$ 0.2        | 95.2 $\pm$ 0.1 |
> | 0.98 | 94.2 $\pm$ 0.2       | 95.1 $\pm$ 0.0        | 95.0 $\pm$ 0.2 |
> | 1    | 94.3 $\pm$ 0.4       | 94.9 $\pm$ 0.2        | 95.6 $\pm$ 0.3 |
> | Ours | **95.0**$\pm$**0.1** | **95.6**$\pm$**0.1**  | **95.9**$\pm$**0.3** |
> |  |  |  |  |
>
> **2. Specifically, the results for $\alpha=1$ and "optimal" $\alpha=1−1/\sqrt{m}$ are quite close, and measured only on 3 datapoints for one dataset. More experimental grounding for motivating the choice $\alpha=1−1/\sqrt{m}$ over $\alpha=1$ would help.**
>
> Although the "optimal choice" looks quite close to $1$ when $m$ is large, the following table, containing the result of $10$ rounds of experiments on a set of simulated data ($2000$ samples generated from uniform distribution 1 in Appendix A.4), shows that even slight changes on $\alpha$ could make the selected samples quite different. The elements in the table is the average cardinality of the set difference between the $m$ samples selected by GKHR with trade-off parameter $\alpha$ and $\alpha_0=1$.
>
> | $\alpha$ \ $m$ | 80 | 160 | 240 | 320 | 400 |
> | ---  | --- | --- | --- | ---  | --- |
> | 0.6        | 36.0000 | 73.1000 | 117.1000 | 163.3000 | 208.0000 |
> | 0.8        | 26.1000 | 53.6000 | 85.8000 | 121.9000 | 160.5000 |
> | 0.9        | 17.7000 | 38.5000 | 63.2000 | 91.9000 | 121.0000 |
> | 0.95       | 15.8000 | 32.4000 | 52.9000 | 77.9000 | 102.8000 |
> |  | | | | | |

---

> ### Author Response · Authors · 2023-11-16
> **Responses To Questions**
>
> **3. Performance under the supervised training paradigm.**
>
> Regarding the query you raised about '*at what subset size the accuracy would match the accuracy of a classifier trained on the whole training set*', we consider this to be a highly valuable research direction. Such comparative experiments would significantly aid in a more comprehensive evaluation of the selection efficiency of our method. We trained classifiers in a supervised manner on CIFAR-10 with annotation budgets of 7,500 (15%), 12,500 (25%), 17500 (35%) and 50,000 (100%), respectively. The table below presents the experimental results. We observed that when the annotation budget is 7,500, our method is comparable to AL methods. Although AL methods also utilize only labeled data for supervised learning, it features a coupling of model training and sample selection, with the model's performance progressively improving during training, thereby tending to select samples more effective for enhancing model performance. Conversely, our approach is agnostic to the data prior to sample selection. With an annotation budget of 12,500, our method's accuracy reached 91.2%. At a budget of 17,500, our method's accuracy attained 93.8%, closely approaching the accuracy of a classifier trained on the whole dataset.
>
> | Methods | Budget | Acc (%) |
> | --- | --- | --- |
> | *Active Learning* |  |  |
> | VAAL | 7500 | 86.8 |
> | UncertainGCN | 7500 | 86.8 |
> | CoreGCN | 7500 | 86.5 |
> | MCDAL | 7500 | 87.2 |
> | *Supervised Learning* |  |  |
> | Ours | 7500 (15%) | 86.6 |
> | Ours | 12500 (25%) | 91.2 |
> | Ours | 17500 (35%) | 93.8 |
> | Whole Dataset | 50000 (100%) | 94.3 |
> | | | |

---

### Official Review · Reviewer_9fK9 · 2023-11-01

**Soundness:** 2 fair
**Presentation:** 3 good
**Contribution:** 2 fair
**Rating:** 5
**Confidence:** 4

**Summary:**

The paper proposes a method for selective a representative as well as a diverse subset for expert annotation. The idea is to weight the terms in MMD-squared distance between the target and the desired subset, such that the trade-off between diversity and representativeness can be explicitly controlled. The weighted distance can be minimized using Frank-Wolfe/Kernel-herding techniques. The kernel-herding algorithm is modified so that samples are not repeated. Error bound for this modified algorithm is presented under certain conditions (Theorem 3.2).

When the proposed criteria is used for AL in context of SSL algorithms, empirically it is shown that the methodology outperforms AL and SSAL baselines. (table 2)

**Strengths:**

1. The idea of \alpha-MMD^2 and its interpretation in (5) are clear and suitable for the AL tasks.

**Weaknesses:**

1. The basic methodology essentially selects a representative and diverse subset. There are many methodologies for such a diverse subset selection. e.g., [1*]-[3*]. None of such methods have been discussed nor have been empirically compared with. This makes it difficult to evaluate the significance of the proposal.

2. The weighting idea and corresponding algorithm details are more or less straightforward. (Mainly because it is a simple modification of MMD).

3. Reg. table2. Since UPA is employed above SSL (flexmatch/freematch), it may be important to compare against baseline/SOTA AL criteria when employed with flexmatch/freematch. Then the advantage of the proposed criteria would be explicit. Now it is not clear whether the imporvement is because of flexmatch/freematch or because of the proposed criteria.



[1*] https://arxiv.org/pdf/2104.12835.pdf
[2*] https://arxiv.org/pdf/1901.01153.pdf
[3*] https://proceedings.neurips.cc/paper/2014/file/8d9a0adb7c204239c9635426f35c9522-Paper.pdf

**Questions:**

1. In the proof of theorem3.2, "j" is defined as argmin over i=1to n\I_p for some objective. But in the proof the following is used: f_{I_p}(x_j_{p+1}) <= f_{I_p}(x_j). Is this correct?

---

> ### Author Response · Authors · 2023-11-16
> **Responses To Weaknesses #1 and #2**
>
> Thank you very much for your comments. We hope our responses can address all your concerns.
>
> **1. None of such methods have been discussed nor have been empirically compared with. This makes it difficult to evaluate the significance of the proposal.**
>
> We appreciate your suggestion regarding the comparative methods. We are currently in the process of replicating these methods, which is expected to take some time. We will provide a response with the results once this task is completed.
>
> **2. The weighting idea and corresponding algorithm details are more or less straightforward (Mainly because it is a simple modification of MMD).**
>
> Firstly, in SSL tasks, especially the procedure of selecting unlabelled data, how to effectively utilize limited labeled data remains an unresolved challenge. Consider MMD is a powerful tool to measuring the distance between distributions, our idea is the first to apply the modification of MMD to resolve such problem via a unique approach.
> Secondly, the reason why we choose MMD is not only because MMD is a proper statistical divergence, but also due to the fact that the modified version of MMD $(\alpha-\operatorname{MMD})$ is capable of measuring both the representativeness and diversity. For other popular statistical divergence, including Wasserstein distance, total variation, etc., they cannot be applied in our task throught such a naturally and succinctly extension. Therefore, our idea is not a simple transplant, but an innovative development that combines the special property of MMD, which prefectly meets the demand of SSL.
> More importantly, the sample selection problem has being neglected for a long time in SSL. In our opinion, this problem is a considerable issue in applications of SSL, especially when labelling is expensive. Our idea provide a new perspective and solution to such problem, and validate the effectiveness of the methodology via sufficient experiments. Our innovative work not only generalizes the concept of MMD, but also contribute a new methodology with theoretical guarantees to the SSL community. We hope that this work attract more attention and interests to the problem of improving the efficiency of data utilization from the community.

---

> ### Author Response · Authors · 2023-11-16
> **Responses To Weaknesses #3 and Questions #1**
>
> **3. Reg. table2. Since UPA is employed above SSL (flexmatch/freematch), it may be important to compare against baseline/SOTA AL criteria when employed with flexmatch/freematch. Then the advantage of the proposed criteria would be explicit. Now it is not clear whether the imporvement is because of flexmatch/freematch or because of the proposed criteria.**
>
> The improvement is related to both aspects. As shown in Table 1 of the manuscript, Flexmatch/Freematch already outperforms the AL/SSAL methods when our method is not used. Furthermore, with the adoption of our method, there is a further enhancement in performance.
>
> In order to provide a more intuitive comparison between our sampling method and existing AL sampling methods, while considering that AL sampling methods only utilize labeled data for supervised training, we devised a new comparative experiment under the same labeling budget constraint. First, we employed our method to select samples for annotation, followed by using these annotated samples for supervised classifier training. This step was undertaken to demonstrate the performance contrast between different sampling methods in supervised learning. The table below presents the experimental results.
>
> | Methods | Budget | Acc (%) |
> | --- | --- | --- |
> | *Active Learning* |  |  |
> | VAAL | 7500 | 86.8 |
> | UncertainGCN | 7500 | 86.8 |
> | CoreGCN | 7500 | 86.5 |
> | MCDAL | 7500 | 87.2 |
> | *Supervised Learning* |  |  |
> | Ours | 7500 | 86.6 |
> ||||
>
> It is worth noting that within AL framework, sample selection is tightly coupled with model training, where model training is dynamic, and subsequent sample selection becomes more precise as performance improves. This is an iterative process that continues until the labeling budget is exhausted. In contrast to our sampling method, all samples that require annotation are determined before training (*i.e.,* in the absence of knowledge about the training data). The challenge with this approach is that all samples must be selected initially, and there is no flexibility to adjust sample selection based on intermediate training results. Although theoretically, this may lead to lower performance compared to dynamic sample selection in AL methods, experimental results demonstrate that even under a static selection framework, our method can achieve performance similar to AL. This indirectly reflects the advantages of our method.
>
> Furthermore, our research aims to enhance the performance of SSL algorithms. We acknowledge that in the SSL domain, the sample selection problem has been a long-neglected and under-researched area. Consequently, there are limited existing methods available for comparison. In our paper, particularly in Table 1, we have conducted a detailed comparison of two commonly used methods in SSL algorithms. The comparative results showcase the superiority of our method, indicating that our approach can improve SSL performance, especially when the annotation budget is limited.
>
> Simultaneously, considering practical application scenarios, especially when dealing with a large amount of unlabeled data and constrained labeling budgets, both AL, SSAL and SSL are viable options. In our paper, Table 2 compares the performance of these different methods in practical applications, aiming to illustrate the variations in results brought about by different choices. We hope this comparison can provide valuable reference and guidance to future researchers facing similar situations.
>
> We hope that our response above addresses your concerns. If you have any further questions, we are open to conducting additional experiments and analyses to provide further clarification.
>
> **4. In the proof of theorem3.2, "j" is defined as argmin over i=1to n\I_p for some objective. But in the proof the following is used: f_{I_p}(x_j_{p+1}) <= f_{I_p}(x_j). Is this correct?**
>
> Thank you very much, again, for pointing out our typo. The inequality is not correct for sure. Actually,
> $$
> \mathbf{e}_j^{\top}\left(\mathbf{K} \mathbf{w}_p-\mathbf{p} _\alpha\right)=f _{\mathcal{I} _p}\left(\mathbf{x} _{j}\right)=\min\limits _{i \in[n] \backslash \mathcal{I} _p} f _{\mathcal{I} _p}\left(\mathbf{x} _i\right)\le f _{\mathcal{I} _p}\left(\mathbf{x} _{j _{p+1}}\right)
> $$
> therefore, the following inequality would be valid (under the assumption in Theorem 3.2).
> $$
> \left(\mathbf{w} _*-\mathbf{e} _j\right)^{\top}\left(\mathbf{K} \mathbf{w} _p-\mathbf{p} _\alpha\right)=\frac{\sum _{i=1}^n f _{\mathcal{I} _p}\left(\mathbf{x} _i\right)}{n}-f _{\mathcal{I} _p}\left(\mathbf{x} _{j}\right)\ge\frac{\sum _{i=1}^n f _{\mathcal{I} _p}\left(\mathbf{x} _i\right)}{n}-f _{\mathcal{I} _p}\left(\mathbf{x} _{j _{p+1}}\right)\ge0
> $$
> we will correct this typo (together with several other typos) in the revised edition of this paper.

---

### Meta-Review · Area_Chair_zqak · 2023-12-03

**Metareview:**

This paper was reviewed by four experts and received 5, 5, 3, 3 as ratings. The reviewers agreed that the paper addresses an important problem, since selecting an optimal subset for annotation is an important problem in scenarios where labels are expensive to get. However, the reviewers raised concerns about the novelty and the experimental evaluation of this paper. Selecting representative and diverse data subsets is a well-researched problem; the proposed method has been compared only against random sampling, which is a very naïve comparison baseline. An exhaustive and thorough comparison with the state-of-the-art methods is lacking and is necessary to understand the merit of the proposed framework. This concern was not addressed convincingly by the authors in the rebuttal.

We appreciate the authors' efforts in meticulously responding to each reviewer’s comments. We also appreciate the authors' efforts in conducting additional experiments to compare their method against several state-of-the-art AL methods, and also their method with different annotation budgets to assess at what subset size the accuracy matches the accuracy of a classifier trained on the full training set. However, during the AC-reviewer discussion period, the reviewers mentioned that the overall approach is still not reinforced enough in the experimental validation, and much more work is needed to publish this method. In light of the above discussions, we conclude that the paper may not be ready for an ICLR  publication in its current form. While the paper clearly has merit, the decision is not to recommend acceptance. The authors are encouraged to consider the reviewers’ comments when revising the paper for submission elsewhere.

**Justification For Why Not Higher Score:**

The reviewers share a common concern that the novelty and experimental validation are weak as selecting diverse samples is a well-studied problem, and the proposed method has not been extensively compared against all the relevant baselines. These do not merit acceptance of this paper at ICLR, considering its high standards.

**Justification For Why Not Lower Score:**

N/A.

---

### Decision · Program_Chairs · 2024-01-16

Reject